# Challenges and Future Trends in Atopic Dermatitis

**DOI:** 10.3390/ijms241411380

**Published:** 2023-07-12

**Authors:** Julius Garcia Gatmaitan, Ji Hyun Lee

**Affiliations:** 1Department of Dermatology, Seoul St. Mary’s Hospital, The Catholic University of Korea, 222, Banpo-daero, Seocho-gu, Seoul 06591, Republic of Korea; juliusggatmaitan@gmail.com; 2Gatmaitan Medical and Skin Center, Baliuag 3006, Bulacan, Philippines; 3Skines Aesthetic and Laser Center, Quezon City 1104, Metro Manila, Philippines

**Keywords:** atopic dermatitis, atopic eczema, biologic therapy, cytokine signaling, treatment

## Abstract

Atopic dermatitis represents a complex and multidimensional interaction that represents potential fields of preventive and therapeutic management. In addition to the treatment armamentarium available for atopic dermatitis, novel drugs targeting significant molecular pathways in atopic dermatitis biologics and small molecules are also being developed given the condition’s complex pathophysiology. While most of the patients are expecting better efficacy and long-term control, the response to these drugs would still depend on numerous factors such as complex genotype, diverse environmental triggers and microbiome-derived signals, and, most importantly, dynamic immune responses. This review article highlights the challenges and the recently developed pharmacological agents in atopic dermatitis based on the molecular pathogenesis of this condition, creating a specific therapeutic approach toward a more personalized medicine.

## 1. Introduction

Atopic dermatitis (AD) is a common chronic inflammatory skin disorder affecting approximately 11 to 20% of children and 5–8% adults. In a retrospective observational study conducted in South Korea by Lee et al., 944,559 pediatric patients and 1,066,453 adults with AD were identified [1,2]. This condition, described as a heterogenous inflammatory skin disease, consists of various subtypes based on clinical (disease stage/chronicity), demographics (age, ethnicity), and molecular parameters/endotypes. The various differences in the AD endotype contributes to this condition’s complexity [3]. While some patients can be managed through conventional therapies, progress toward an innovative therapeutic approach based on AD subtype is essential to achieve the therapeutic benefits the patient expect. The recurrent eczematous lesions and intense pruritus severely affect the patient’s quality of life and, therefore, have a significant negative impact on their psychological and social well-being. A national, multicenter, non-interventional study in South Korea revealed that out of 1163 patients, the quality of life of 72.3% (840) patients was identified to be moderately to severely affected, indicating the high disease burden of this condition [2]. The mainstay of therapy in AD is topical agents. In patients with mild to moderate AD, the standard medical treatment is anti-inflammatory drugs such as topical corticosteroids and topical calcineurin inhibitors. If the patients cannot achieve disease control and clinical improvement despite the topical therapies, systemic treatments including phototherapy sessions are recommended. If topical agents like emollients, corticosteroids, and calcineurin inhibitors with or without phototherapy still fail to control the clinical condition of the patient, systemic immunomodulatory agents such as cyclosporine, methotrexate, azathioprine, mycophenolate mofetil, and short-term systemic corticosteroids are recommended. Avoidance of triggering factors that exacerbate pruritus such as woolen clothes, emotional stress, and uncomfortable climatic conditions is essential at all times [4]. In addition to the treatment armamentarium available for AD, novel drugs targeting significant molecular pathways in AD such as biologics and small molecules are also introduced given the condition’s complex pathophysiology [5]. Among the challenges faced in this condition are the following: trigger avoidance, treatment adherence due to adverse effects of topical steroids and systemic immunosuppressant agents, and more importantly, the economic burden on some patients due to the cost of biologics and JAK/STAT inhibitors. Like any other drug, these newer biologics and small molecules are not a one-size-fits-all. While most of the patients are expecting better efficacy and long-term control, the response to these drugs would still be influenced by numerous factors such as complex genotype, diverse environmental triggers and microbiome-derived signals, and, most importantly, dynamic immune responses [6]. This review article highlights the challenges and the recently developed pharmacological agents in AD based on the pathogenesis and provides an evidence-based treatment approach toward more personalized medicine.

## 2. Current and Future Molecular Targets in Atopic Dermatitis

AD is a skin condition with a complex pathogenesis. Targeting the specific pathophysiology through modulation of the skin microbiome and improvement of the skin barrier in combination with the selection of appropriate agents that target the innate and adaptive immune system would significantly make an impact in improving the itch–scratch cycle. This review article is divided in five sub-categories:

### 2.1. Modulation of the Skin Microbiome

Numerous studies have shown the important role of the skin’s microbial populations in the promotion of T-cell-driven inflammation in AD. The increase in *Staphylococcus aureus* colonization, as seen in Figure 1a,b, influences a decrease in other commensal bacterial communities, which are necessary to produce antimicrobial peptides (AMPs) to prevent overgrowth of pathogenic *Staphyloccus aureus*. AMPs are produced by the skin to inhibit or destroy the growth of microbes. The human body consists of over 20 peptides, those of particular importance in AD of which are cathelicidin, human β-defensin-1, dermcidin, and ribonucleases. The modification in the expression and secretion of AMP may increase the susceptibility of the skin of AD patients to infections by bacteria, viruses, and fungi.

Weiss et al. showed that the application of topical niclosamide (ATx201) (Figure 1b) resulted in a decrease in *Staphylococcus aureus* colonization and an augmentation in the Shannon diversity of the skin microbiome among patients with AD. In their randomized double-blind placebo-controlled Phase 2 trial, topical niclosamide qualitatively improved the AD lesions of 36 adults for 7 days in a twice-a-day treatment. Topical ATx201 has also been found to be less resistant against *Staphylococcus aureus* as compared with fusidic acid cream, hence being well suited for longer-term application [7].

In a phase 2 randomized controlled trial investigating 80 adult participants with mild to moderate AD, topical antimicrobial peptide omiganan (Figure 1b) in various concentrations (1%, 1.75%, or 2.5%) or a vehicle was applied topically twice a day for 28 days on all lesions. While topical administration led to a recovery of dysbiosis, no significant clinical improvement in AD lesions was observed [8]. A recent study by Javia et al. demonstrated a reduction in pro-inflammatory cytokines and improvement of clinical lesions on an AD and psoriasis mice model using the omiganan liposomal formulation versus omiganan gel and lotion [9].

*Roseomonas mucosa*, a skin commensal, inhibited the growth of *Staphylococcus aureus* in in vitro studies [10]. Zeldin et al. demonstrated that *Roseomonas mucosa* fixes the nitrogen essential to the production of protective glycolipids and ceramides. In their phase 2 randomized double-blind placebo-controlled multicenter trial investigating 154 children, adolescent, and adult subjects (aged 2 years and older) with mild to moderate AD, topical FB-401 (*Roseomonas mucosa*) (Figure 1b) (1%, 1.75%, or 2.5%) or placebo was applied topically thrice a week for 16 weeks on all lesions. While FB-401 failed to meet the end points in the placebo-controlled trial, the subgroup who completed the full protocol demonstrated clinically modest and statistically significant clinical improvements [11].

Topical *Staphylococcus hominis* A9 (*Sh*A9) (Figure 1b), a lyophilized strain with a dual activity of killing *Staphylococcus aureus* and inhibiting the production of *S. aureus*-derived toxins, allowed the recovery of the skin microbiome in the study of Nakatsuji et al. [12]. A phase 1 open-label study investigating the pharmacokinetics of topical Sha9 on the skin of patients with AD after single application for 24 days in 20 participants with moderate to severe AD is currently in their recruitment phase with an estimated study completion date of September 2023 [13].

B244 consists of a purified strain of *Nitrosomonas eutropha*, an ammonia oxidizing Gram-negative bacteria (AOB) isolated from soil samples. The organism diminishes the survival of *Staphylococcus aureus* (Figure 1b) and exhibits an in vitro reduction in Th2-associated cytokines such as IL-4, IL-5, and IL-13. The primary endpoint of their study was the mean change in WI-NRS from baseline to week 4. A randomized, placebo-controlled, double-blind phase 2b trial was conducted on adults aged 18–65 years with mild-to- moderate AD and moderate-to-severe pruritus. Participants were randomized into low-dose, high-dose, or a vehicle group for the 4-week treatment period and a 4-week follow-up period. Participants were then instructed to apply the topical spray twice daily throughout the treatment period. In their study, the researchers were able to find a reduction of 34% (−2.8 B244 vs. −2.1 placebo, *p* = 0.014 and *p* = 0.015 for the high-dose and low-dose, respectively) from a baseline score of >8. B244 was well tolerated without serious adverse events (SAEs), treatment-emergent adverse events (TEAEs), and treatment-related TEAEs. Based on the findings of their study, the drug may be promising novel, natural, fast-acting topical spray treatment option for AD and associated pruritus [14].

Hence, the microbiome has become a major research interest because of the interaction between the immune system and the skin condition. Future strategies based on recent prophylactic and therapeutic approaches will be directed toward the diversity of the skin microbiomes, as seen in Figure 1b. Therapeutic manipulation of the gut microbiota using prebiotics, probiotics, symbiotics, and postbiotics is beyond the scope of this review article.

### 2.2. Targeting the Innate Immune System

#### 2.2.1. Innate lymphoid Cells (ILCs)

ILCs functionally resemble T-cells but lack the clonal antigen receptors. These cells require cytokines instead of antigens [15]. Group 2 innate lymphoid cells (ILC2) belong to the innate immune system leukocytes known to secrete pro-allergic cytokines such as IL-4, IL-5, IL-9, and IL-13, as depicted in Figure 1a, and are therefore implicated in initiating a Th2 response. Th2 responses with the overexpression of IL-4, IL-5, IL-9, and IL-13 are found increased in both extrinsic and intrinsic AD patients [16]. ILC2 can be activated by interleukin 25 (IL-25), interleukin-33 (IL-33), and thymic stromal lymphoprotein (TSLP) or eicosanoids without antigen stimulation. In a steady state, the presence of ILC2 is unique in the skin, keratoconjunctiva, lungs, and intestines. When inflammation occurs, there is discharge of the activated sentinel ILC2s from the skin to the circulatory system like the lymph nodes and peripheral blood [17]. Activation of a local innate response such as ILC2 occurs first, followed by the activation of adaptive immunity such as Th2 cells [18,19]. Translating these findings by Leyva and Dominguez clinically, in order to prevent the transition of infants with AD to food allergy, asthma, and allergic rhinitis, eczematous lesions must be strongly controlled [15].

#### 2.2.2. Alarmins

##### Thymic Stromal Lymphopoietin (TSLP)

Keratinocytes, fibroblasts, dendritic cells, stromal cells, and mast cells produce TSLP, as indicated in Figure 1a, to induce a Th2-type immune response, triggered by environmental stimuli like allergens and imbalances in the skin barrier dysfunction [20]. The TSLP serum level in the skin of AD patients was found to be elevated in the study of Lee et al. [21].

In a phase 2a trial, the efficacy of subcutaneous tezepelumab, an anti-TSLP monoclonal antibody (Figure 1b) administered in 280 mg, in combination with topical corticosteroids every 2 weeks for 12 weeks, was investigated. However, the study findings indicated a lack of statistically significant improvement in EASI-50, EASI-75, and the pruritus numeric rating scale among patients with moderate-to-severe AD [21]. While the drug showed a non-satisfactory outcome in AD, tezepelumab injection was approved as an add-on maintenance treatment to improve severe asthma [22].

##### Interleukin-1 (IL-1) Family

The interleukin-1 family functions to regulate the immune response and link the innate and adaptive responses. This family is composed of 11 cytokine members, seven agonists (IL-1α, IL-1β, IL-18, IL-33, IL-36α, IL-36β, and IL-36γ), and four antagonists (IL-1Ra, IL-36Ra, IL-37, and IL-38) [23]. Based on their structural and functional characteristics, the family is further subdivided into subfamilies (IL-1, IL-18, IL-33, and IL-36), with each one having a specific receptor, respectively (IL-1R1, IL-18Rα, IL-33R, and IL-36R). MyD88 is a central signal adaptor involved in the signal transduction of the cytokines of the Interleukin-1 family. Didovic et al. were able to demonstrate its role in the generation of an adaptive and immune response in skin inflammation in an AD-like murine study [24].

-Interleukin-1 subfamily

Interleukin 1 (IL-1)

IL-1α and IL-1β are both pro-inflammatory cytokines (Figure 2a) that play a significant role in the innate immune response. IL-1α is expressed in cutaneous keratinocytes while IL-1β is inducible in monocytes, macrophages, and dendritic cells. The binding of either cytokine, specifically IL-1α, to the IL-1R1 receptor (IL-1R1) subsequently recruits interleukin-1 receptor accessory protein (IL-1RAcP), also called IL-1R3, which promotes an inflammatory gene expression targeting either type 1 or type 17 immune responses, as presented in Figure 2b [25]. On the other hand, Interleukin-1Rα (IL-1Rα), the antagonist of IL-1α and IL-1β, is responsible for anti-inflammatory effects after binding to the IL-1R1 receptor (Figure 2a) [26]. While much evidence supports the role of IL-1 cytokines in the development of the acute and chronic phase of AD, the results targeting the pro-inflammatory cytokines such as IL-1α and IL-1β are still being investigated.

Bermekimab, a fully human monoclonal antibody that targets IL-1α (Figure 2b), demonstrated significant and encouraging results in patients with hidradenitis suppurativa and psoriasis [27,28]. A phase 2b, multicenter, randomized, placebo- and active-comparator-controlled, double-blind study that evaluated the safety and efficacy of subcutaneous bermekimab in adult (18 years and older) participants with moderate-to-severe AD was terminated due to the lack of efficacy (GENESIS) [29].

Canakinumab, a fully human monoclonal antibody against IL-1β (Figure 2b), has shown promising clinical efficacy and safety in various inflammatory diseases like cryopyrin-associated periodic syndromes (CAPS) and possibly other complex inflammatory diseases, such as rheumatoid arthritis, systemic-onset juvenile idiopathic arthritis (SoJIA), chronic obstructive pulmonary disease, and ocular diseases [30]. Schwartz et al. demonstrated in their study that chronic dermatitis in AD-induced mice developed independently of IL-1α but was heavily dependent on IL-1β instead. The study was also able to reveal that treatment with anti-IL-1β-antibody alleviated the exacerbation of dermatitis [31]. The current evidence of canakinumab is uncertain but targeting this specific interleukin may be a potential treatment in AD.

Gevokizumab, a neutralizing humanized monoclonal antibody specific to IL-1β (Figure 2b), given as 60 mg subcutaneously every 4 weeks for 12 weeks, showed its efficacy in a case study of two patients with generalized pustular psoriasis, which resulted in a decrease in generalized pustular psoriasis area and severity index (GPPASI) scores by reductions of 79% and 65% at weeks 4 and 12, respectively [32]. Its potential role in AD is yet to be determined.

Interleukin-36 subfamily (IL-36)

The cytokines are composed of three agonists (IL-36α, IL-36β, and IL-36γ) and two antagonists (IL-36 receptor antagonist (IL36Ra) and IL-38) [33]. These cytokines bind to the ligand-binding IL-1R6 receptor and recruits 1L1-1R3 (accessory protein common to other receptor complexes) as a coreceptor, signaling transduction to the nucleus and the transcription of inflammatory genes, as visualized in Figure 2a [34]. HaCat cells treated with IL-36 inhibited keratinocyte differentiation and led to filaggrin downregulation in a study by Wang et al., which might suggest its involvement in barrier dysfunction [35]. In the study of Patrick et al., the group found increased serum levels of IL-36α and IL-36γ along with elevated serum IgE in AD patients. In addition, increased IL-36α and IL-36γ expressions in keratinocytes in the histologic sections of inflamed skin were also observed [36]. An in vivo and vitro study by Liu et al. demonstrated the role of the IL-36-mediated T cell response following epicutaneous exposure to *Staphylococcus aureus* [37].

Bissonnette et al. conducted a phase 2a proof-of-concept study involving 51 adult patients with moderate-to-severe AD to evaluate the effectiveness of spesolimab, an anti-IL36R antibody (Figure 2b). While the intravenous administration of spesolimab at a dose of 600 mg every 4 weeks was found to be safe and well tolerated, there was no statistically significant difference in the percent change from baseline EASI up to week 16 compared to the placebo group. As a result, the researchers concluded that the IL-36 pathway does not have a major role in the pathogenesis of AD, explaining the modest effect of this drug compared with placebo [38].

2.Interleukin-18 subfamily (IL-18)

Interleukin-18 (IL-18) plays an essential role in the regulation of the innate and acquired immune response against pathogens [39]. Active IL-18 cytokine binds to the interleukin-18 receptor alpha (1L18Rα) receptor and recruits interleukin-18 receptor beta (1L18Rβ) to stimulate the production of IFN-γ and can induce both Th1 and Th2 cellular responses [40]. Interleukin-18 binding protein (IL18-BP) regulates its pro-inflammatory activity by creating a negative feedback mechanism by sequestering the cytokine Interleukin 37 (IL-37) to bind to interleukin-18 receptor alpha (1L18Rα) and restrict the interleukin-18 receptor (1L-18R)-dependent inflammation and inhibiting its pro-inflammatory cytokine production [25]. IL-18 is released not only by keratinocytes but inflammatory dendritic epidermal cells as well [41]. Based on the presence or absence of interleukin-12 (IL-12), IL-18 can stimulate both Th2 and Th1 responses due to its pleiotropic effect. Exposure to allergens such as house dust mites and to lesional infections induces the expression of IL-18 [40]. The action of the IL-18 subfamily is depicted in Figure 2a. Ohnishi et al. also found that the plasma IL-18 level was elevated in children with AD and showed a positive correlation with the clinical severity of AD [42]. These cytokines bind to the ligand binding IL-1R6 receptor and recruits 1L1-1R3 (accessory protein common to other receptor complexes) as the coreceptor, signaling transduction to the nucleus and the transcription of inflammatory genes [34]. Chymase, an enzyme released by mast cells through activation of IL-18, might contribute to the inflammatory responses in AD lesions [40].

Tadekinig alfa is a human recombinant IL-18-binding protein (Figure 2b) that is being investigated in a phase II open-label clinical trial on patients with adult onset Still disease [43]. Therefore, downregulating IL-18 activity seems to be a future approach for treatment of AD.

##### Interleukin 33 (IL-33)

IL-33 binds to the Interleukin-33-specific receptor (ST2) and recruits interleukin-1 receptor accessory protein (IL-1RAcP) as the coreceptor, signaling transduction to the nucleus and the transcription of inflammatory genes promoting Th2-type immunity, as depicted in Figure 2a [44]. Considered to be a “danger alarmin”, this cytokine detects tissue damage to local immune cells and activates the innate immune system such as ILC2, mast cells, and basophils to secrete type 2 cytokines, chemokines, and pro-inflammatory mediators, as visualized in Figure 1a [45]. Upon the activation of this cytokine, the dermal basophils are stimulated to produce interleukin-4, which may contribute to the inflammation process [46]. The expression of IL-33 in the epidermis of patients with AD correlated with EASI and SCORAD in a study by Guttman-Yassky et al. [47]. In another study, Tamagawa-Mineoka et al. assessed the serum levels of IL-33 in individuals with AD, urticaria, and psoriasis, comparing them to those of a healthy control group. The researchers concluded that IL-33 released from skin with disrupted barriers could contribute to the inflammation associated with AD, making it a valuable marker for monitoring the effectiveness of drug therapy [15,48].

Etokimab is a humanized anti-IL-33 monoclonal antibody (Figure 2b). A phase 2a proof-of-concept study of a single intravenous 300 mg dose of etokimab in 12 adult patients with moderate-to-severe AD was able to demonstrate an EASI-50 on 83.3% of patients and an EASI-75 on 33% of patients after 29 days of etokimab administration [49]. A phase 2b randomized, double-blinded, placebo-controlled, multi-dose study in approximately 300 adults with moderate-to-severe AD failed to demonstrate a statistically greater improvement in EASI compared with placebo at week 16 [50].

Although this molecule is a good biomarker, a study assessing the efficacy, safety, and pharmacokinetic profile of itepekimab (REGN3500), an anti-IL33 antibody (Figure 2b), administered subcutaneously to adult patients with moderate-to-severe AD failed to reach significance in a phase 2b clinical trial [51].

Astegolimab, a fully human IgG2 monoclonal antibody that binds to ST2 (Figure 2b) and inhibits further signaling, given as 490 mg every 4 weeks for 16 weeks, was not able to demonstrate a significant difference compared with placebo for the primary outcome (change in EASI from baseline to week 16) and secondary outcomes (blood eosinophils, serum IL-5, and serum CCL13) in a phase 2 randomized clinical trial in 65 patients with AD by Maurer et al. [52].

A study on the efficacy and safety of subcutaneous MEDI3506 (MEDI3506 dose 1, MEDI3506 Dose 2, MEDI3506 dose 3, or placebo) for 16 weeks in 148 adult subjects with moderate-to-severe AD has already been completed, but the results are not yet available [53]. In a study by England et al., the group was able to demonstrate the dual pharmacology of tozorakimab (MEDI3506) by inhibiting IL33 signaling via IL33R/ST2 (Figure 2b) and RAGE/EGFT in reducing inflammation and epithelial dysfunction [54].

Though numerous clinical studies were conducted, the challenge in targeting the IL-33/ST2 signaling pathway in AD may not be sufficient, due to the complexity of the pathogenesis of this condition.

#### 2.2.3. Aryl-Hydrocarbon Receptor (AHR)

AHR is a ligand-dependent transcriptional factor that can be found on the cytosol in complex with aryl hydrocarbon interacting protein (AIP), prostaglandin E synthase 3, and heat-shock protein (HSP) 90 kda. After binding, the AHR translocates to the nucleus and binds to DNA to regulate target gene expression [55]. These receptors are expressed in immune and non-immune cells and, once activated, can exert both pro- and anti-inflammatory activities depending on various exogenous and endogenous factors. In vitro and in vivo studies have shown that AHR is a ubiquitous ligand (Figure 1a and Figure 3a) and can be found on T regulatory cells (Treg), Type 1 regulatory cells (TR1), T-helper cells 17 (Th17), T-helper cells 22 (Th22), cytotoxic effector cells (CD8+), Gamma delta (*γδ*) T cells, and innate lymphoid cells (ILC-1, ILC-2, ILC-3) [55].

Tapinarof, a topical AHR agonist (Figure 1b and Figure 3b), has been approved by the FDA for psoriasis. As of this writing, an open-label long-term extension study is currently recruiting to evaluate the safety and efficacy of tapinarof cream 1% in 961 participants (>2 years old) with AD. This study will consist of up to 48 weeks of treatment and a 1-week safety follow-up period. The estimated study completion date is September 2024 [56].

Due to the expression of AHR in various cells, this receptor represents a promising target for therapy for inflammatory skin disorders.

### 2.3. Targeting the Adaptive Immune System

#### 2.3.1. Antigen Presentation through OX40-OX40L

As an essential costimulatory T-cell receptor molecule and member of the TNF superfamily, OX40 is primarily expressed on enhanced effector T-cells (Th1, Th2, Th17, and Th22) and regulatory T-cells [57]. OX40 is essential for T-cell expansion, survival, and memory differentiation. Conversely, the ligand of OX40, known as OX40L, is expressed on activated antigen-presenting cells (such as dendritic cells and macrophages) as well as endothelial cells [58]. Pro-inflammatory T-cell responses and T-helper memory cells generation are achieved through the engagement and interaction of OX40 with OX40L, as demonstrated in Figure 1a. This pathway is crucial for sustaining the T-cell effector function of the Th2 and Th1 subpopulation. In addition to its vital role in T cells, OX40/OX40L signaling also plays a role in the differentiation and maturation of dendritic cells (DCs). Immature human DCs do not express OX40L initially, but its expression is rapidly induced after stimulation with the soluble form of CD40L (sCD40L). Engagement of OX40L leads to an increase in the expression of CD80, CD86, CD54, and CD40 on mononuclear-derived DCs. Furthermore, OX40L can significantly enhance the secretion of IL-4, IL-6, IL-12, TNF-α, and IL-1β by 4- to 35-fold. These findings suggest that the reverse signaling of OX40L promotes the maturation of dendritic cells [59]. Therefore, targeting this pathway may prove to be a potential therapeutic treatment in AD.

Guttman-Yassky et al. demonstrated the potential of targeting OX40 in patients with AD using telazorlimab (GBR830), a humanized monoclonal anti-OX40 antibody (Figure 1b). In their repeated-dose phase 2a study, which was randomized, double-blind, and placebo-controlled, two doses of 10 mg/kg of intravenous GBR830 were administered to 34 adult participants with moderate-to-severe AD, with a 4-week interval between doses. The treatment was well tolerated and resulted in significant progressive reductions in epidermal hyperplasia and the expression of Th1, Th2, and Th17/Th22 in the lesional skin. These effects were observed until day 71, which was 42 days after the last dose. During the final skin biopsy assessment visit, EASI50 was achieved in 76.9% of GBR-830-treated participants versus 37.5% of placebo-treated patients [60]. In a randomized, multinational, double-blind, placebo-controlled, phase 2b study, Sher et al. investigated the efficacy, safety, pharmacokinetics, and pharmacodynamics of different subcutaneous regimens of telazorlimab (GBR830) in 462 adult patients with moderate-to-severe AD. The study was divided into two parts. For part 1, 313 adults were randomized (1:1:1:1) to telazorlimab 300 mg every 2 weeks, telazorlimab 300 mg every 4 weeks, telazorlimab 75 mg every 4 weeks, or placebo. For part 2, 149 adults were randomized (1:1) to telazorlimab 600 mg every 2 weeks versus placebo. The primary endpoint of the study was the percent change from baseline in the EASI score at week 16. The group was able to conclude that for both part 1 and part 2, the EASI score at week 16 was met by the highest dose of telazorlimab. TEAEs were comparable for both telazorlimab and placebo in both part 1 (65.4% vs. 72.5%) and part 2 (65.3% vs. 50.0%) of their study, with atopic dermatitis, nasopharyngitis, respiratory tract infections, and headache being the most commonly reported. In their conclusion, telazorlimab had a favorable safety and tolerability profile and was able to improve clinical signs and symptoms of moderate-to-severe AD [61].

Rocatinlimab, a fully human immunoglobulin G1 anti-OX40 monoclonal antibody (Figure 1b) administered subcutaneously, holds promise as a treatment for moderate-to-severe AD. In a double-blind, placebo-controlled phase 2 trial, rocatinlimab (KHK4083) was given subcutaneously at various doses (300 mg or 600 mg every 2 weeks, or 150 mg or 600 mg every 4 weeks) versus placebo for 36 weeks. The study included 267 adults with moderate to severe AD who had insufficient control with topical agents. The results showed that rocatinlimab was both safe and effective. Notably, there was an improvement in skin conditions and a reduction in mean serum IgE concentrations from week 16 to week 36. Patient follow-up extended until week 56, revealing sustained improvements even after discontinuation at week 36 [62]. Currently, a phase 3, 24-week, randomized, placebo-controlled double-blind trial [ROCKET-IGNITE] is recruiting 700 adult participants with moderate-to-severe AD to further evaluate two different doses (rocatinlimab dose 1 every 4 weeks (Q4W) + loading dose at week 2 and rocatinlimab dose 2 Q4W + loading dose at week 2) versus placebo (placebo Q4W+ loading dose at week 2). The study aims to achieve a validated Investigator’s Global Assessment for Atopic Dermatitis (vIGA-AD) score of 0 or 1 with a ≥2 point reduction from baseline and ≥75% reduction from baseline in EASI score at Week 24 [63].

Amlitelimab, a monoclonal antibody of the non-depleting IgG4 class targeting OX40L (Figure 1b), effectively inhibits T-cell activation on antigen-presenting cells. Furthermore, it also blocks the pro-inflammatory activity of T-cell-independent antigens by interrupting OX40L reverse signaling [64,65]. In a phase 2a study involving 89 adult patients with moderate-to-severe AD, a multi-center, parallel-group, double-blind, randomized, placebo-controlled study was investigated. The participants were randomly assigned to receive a loading dose of amlitelimab at 200 mg followed by 100 mg every 4 weeks (low dose), a loading dose of amlitelimab at 500 mg followed by 250 mg every 4 weeks (high dose), or placebo. The treatment period lasted for 12 weeks, followed by a 24-week follow-up period. The primary outcome measure was the percentage change in Eczema Area and Severity Index (%EASI) from baseline to week 16. The study demonstrated that both the low dose and high dose of amlitelimab achieved baseline EASI scores at week 16 compared to the placebo group. Out of the 88 participants, 59 were evaluated at week 16. The mean percentage change in EASI from baseline at week 16 was 80.12% for the low dose and 69.97% for the high dose of amlitelimab, compared to 49.37% for the placebo. Notably, a response to treatment was observed as early as week 2. Safety follow-up was conducted until week 36, and both doses of amlitelimab were found to be well tolerated [66]. An ongoing phase 2 long-term extension study is currently underway to assess the safety, tolerability, and efficacy of subcutaneous amlitelimab in 300 adult participants with moderate-to-severe AD. These participants were previously enrolled in the KY1005-CT05 (DRI17366) study and are being followed-up as per the study protocol. The primary objective of this extension study is to determine the percentage of participants who experience treatment-emergent adverse events (TEAEs) from baseline up to week 120 [67]. Notably, the inhibition of the OX40-OX40L pathway does not impact naïve and resting memory T-cells, suggesting it as a safe potential option for the treatment of AD in future studies.

#### 2.3.2. T-Helper-2-Related Cytokines

##### Interleukin-4 (IL-4) through Il-4Rα Receptor

Th-2 cytokines, IL-4, and IL-13 are important for chronic pruritus [68] and can exert their function by binding to the IL-4 receptor (IL4R). There are two types of IL4R complex according to the components of the heterodimeric protein subunit. IL-4Rα is one subunit of IL4R complex with a specific binding affinity for IL-4. The first type is the Type I IL4R, which is composed of IL-4Rα and γc subunits, and the second type is the Type II IL4R composed of IL-4Rα and IL-13Rαl subunits. IL4 cytokine can either bind to Type I IL4R or Type II IL4R to exert its effect, as illustrated in Figure 3a [69,70,71]. Therefore, based on these studies, it is safe to conclude that the type II IL4R complex serves as a functional receptor for both IL-4 and IL-13 [72].

Dupilumab, a fully humanized monoclonal antibody targeting IL4Rα (Figure 3b), is recognized as a primary treatment option for severe, recurrent, or refractory AD. This medication has obtained FDA approval for the management of moderate to severe AD in children (6 months to 6 years), adolescents (6–17 years), and adults [5].

CBP201 is a human IgG4 monoclonal antibody that binds to IL-4Rα (Figure 3b) administered subcutaneously or intravenously. A post hoc analysis on a phase 2b, randomized, double-blind, placebo-controlled trial investigated the safety and efficacy of CBP201 in rapid and sustained improvement across all four (head and neck, trunk, upper and lower limb) body regions among 255 adults with moderate-to-severe AD. This study followed a larger, global phase trial that already achieved its primary and secondary endpoints. The dosing for the treatment group was 600 mg loading followed by 300 mg every 2 weeks versus placebo for 16 weeks. The investigators reported that the baseline EASI was 26.9 and 54.5% had an IGA of 4. The IGA response in CBP201 was highly significant compared with placebo at week 16. A total of 30.3% of study participants in the CBP201 arm compared with 7.5% of study participants with placebo achieved IGA 0–1 and had a reduction of ≥2-points at week 16. Significant improvements in EASI-50, EASI-75, and EASI-90 at week 16 were observed with CBP201 compared with placebo. Significant improvements in pruritus were noted with CBP-201 as early as week 1. The group was able to conclude that CBP-201 was efficacious, safe, and well tolerated. Responses were rapid and sustained through to week 16 [73].

Currently, a phase 2 multi-center, randomized, placebo-controlled, double-blind, parallel-group, dose-ranging study to evaluate the efficacy and safety of AK120 (antibody against IL-4Rα) (Figure 3b) subcutaneous injection every 2 weeks for 30 weeks (regimen 1 and regimen 2) versus placebo in 105 adult subjects with moderate-to-severe AD is recruiting subjects in their study and is estimated to be completed in January 2024 [74].

A multi-center, randomized, double-blind, placebo-controlled phase IIb was conducted to evaluate the efficacy, safety, pharmacodynamics, pharmacokinetics, and immunogenicity of subcutaneous CM310, an anti-IL-4Rα monoclonal antibody (Figure 3b), for adults with moderate-severe AD subjects. The study consisted of three periods, an up-to-4-week screening period, a 16-week randomized treatment period, and an 8-week safety follow-up period. It also included the high-dose arm (600 mg for 1st dose, and then 300 mg for 2nd to 8th doses, every 2 weeks), the low-dose arm (300 mg for 1st dose, and then 150 mg for 2nd to 8th doses, every 2 weeks), and placebo. The primary outcome measure was EASI-75 at week 16. Their results showed that EASI-75 was achieved in 73.1% in the high-dose group versus 70.6% in the low-dose group versus 18.2% in the placebo group. Currently, a phase III study is being conducted (NCT05265923, NCT04893707) [75].

##### Interleukin-13 (IL-13)

IL-13 has garnered significant attention as a potential and valid target in the context of AD, owing to its relevance in the skin condition. The binding of either IL-4 or IL-13 to the IL-13Rα1 subunit of the Type II IL4R triggers the recruitment of IL-4Rα, subsequently activating Janus kinase 1 (JAK1) and tyrosine kinase 2 (TYK2). This activation leads to the phosphorylation of signal transducer and activator of transcription 6 (STAT6), which promotes the polarization of T-helper cells toward the Th2 phenotype. Additionally, IL-13 has the ability to bind to a decoy receptor known as IL-13Rα2, which has been studied for its involvement in collagen deposition, the itch–scratch cycle, fibrotic tissue remodeling, and the endogenous regulation of IL-13 [76]. The action of IL-13 can be summarized in Figure 3a. Among the pathogenic factors of IL-13 are the following: keratinocyte alteration, reduction in antimicrobial peptide production, promotion of eosinophil recruitment, and stimulation of IgE [76].

Tralokinumab, a fully humanized IgG4λ, binds to IL-13 at an epitope that overlaps with the binding site of receptors IL-13Rαl subunit and IL-13Rα2 subunit, thereby preventing its interaction and blocking further signaling (Figure 3b) [77]. This drug has obtained approval in the United States and Europe for the treatment of moderate-to-severe AD in adults. Currently, a randomized, parallel-group, monotherapy trial to assess the pharmacokinetics and safety of tralokinumab in children (2 to ≤12 years old) with moderate-to-severe AD is being studied. The aim is to recruit participants and gather data until the estimated completion date of 30 September 2025 [78].

Cendakimab (RPC-4046), a recombinant humanized anti-interleukin 13 monoclonal antibody, binds to IL-13 at an epitope that overlaps with the binding site of receptors IL-13Rαl subunit and IL-13Rα2 subunit, thereby preventing its interaction and blocking further signaling (Figure 3b) [77]. This drug is currently under development for the treatment of eosinophilic esophagitis [79], eosinophilic gastroenteritis [80], and AD [81]. A phase 2, multicenter, global, randomized, double-blind, placebo-controlled, parallel-group study evaluating the safety and efficacy of cendakimab (CC-93538) given subcutaneously in three different doses versus placebo in 214 adults with moderate-to-severe AD was completed in November 9, 2022 and the results are still being completed as of writing [81].

Lebrikizumab, a humanized monoclonal IgG4 antibody that selectively blocks IL13 and prevents the heterodimerization of the type II IL4R receptor composed of IL4Rα and IL13Rαl subunits (Figure 3b) [77], already established its safety and efficacy in AD. In two identical 52-week phase 3 trials by Silverberg et al., lebrikizumab demonstrated significant efficacy in treating moderate-to-severe AD. Adolescents (12 to <18 years) and adults (≥18 years) received subcutaneous lebrikizumab 250 mg every 2 weeks (with a loading dose of 500 mg). Compared to placebo, lebrikizumab achieved higher rates of primary outcomes: IGA score of 0 or 1 with a reduction of ≥2 points from baseline at week 16 (43.1% and 33.2% vs. 12.7% and 10.8% for trial 1 and trial 2, respectively), EASI-75 response at week 16 (58.8% and 52.1% vs. 16.2% and 18.1% for trial 1 and trial 2, respectively), and EASI-90 response at week 16 (38.3% and 30.7% vs. 9.0% and 9.5% for trial 1 and trial 2, respectively) [82]. Blauvelt et al. conducted a 52-week study to assess the efficacy and safety of lebrikizumab monotherapy in patients with moderate-to-severe AD. After the treatment period, a significant proportion of patients maintained positive outcomes. Patients receiving lebrikizumab every 2 weeks demonstrated the maintenance of IGA scores of 0 or 1 with a reduction of ≥2 points in 71.2%, while the EASI-75 response was maintained at 78.4%. For patients receiving lebrikizumab every 4 weeks, 76.9% had an IGA score and 81.7% had an EASI-75 response. Lebrikizumab withdrawal showed slightly lower maintenance rates, with 47.9% for IGA score and 66.4% for EASI-75 response. Overall, the study concluded that lebrikizumab every 2 weeks, as well as every 4 weeks, maintained similar improvements in the signs and symptoms of moderate-to-severe AD following a 16-week induction period [83].

Eblasakimab is a fully humanized monoclonal antibody that targets the IL-13Rα1 subunit (Figure 3b), therefore inhibiting the signals by IL-4 and IL-13, showing great promise in the treatment of AD. A randomized, double-blinded, placebo-controlled, proof-of-concept study on eblasakimab given subcutaneously once a week in three different doses (200 mg, 400 mg, or 600 mg) versus placebo in 49 adults with moderate-to-severe AD for 8 weeks was completed in December 2022. The study demonstrated that EASI-75 was achieved in 13%, 50%, 57%, and 50% in the placebo, 200 mg group, 400 mg group, and 600 mg group, respectively. A decrease in the peak pruritus NRS by 37% in the 600 mg groups compared with a 16% decrease in the placebo group was statistically significant [76]. Currently, a randomized, double-blind, placebo-controlled, multicenter trial to evaluate the efficacy and safety of eblasakimab in 75 adult participants with moderate-to-severe AD previously treated with dupilumab is being conducted. The study aims to compare ASLAN004 (eblasakimab) given at a loading dose of 600 mg at week 0 and week 1, and 400 mg every week from week 2 to week 15 compared with placebo. The primary outcome is the percent change from baseline in EASI at week 16. The study is currently in the recruitment phase and the estimated study completion date is 16 December 2023 [84].

##### Interleukin-5 (IL-5) through IL-5rα Receptor

Interleukin 5 cytokine binds to interleukin 5-receptor (IL-5R) consisting of IL-5 receptor alpha subunit (IL5Rα) and a common receptor B subunit (BC), as depicted in Figure 3a. IL-5 specifically binds to the IL5Rα and recruits the beta chain to activate the downstream signaling molecules for the stimulation of B-cells and eosinophils [85]. IL-5 plays the most significant role in eosinophil biology, is detectable in the inflammatory infiltrate of AD, and correlates well with the disease severity [6].

Mepolizumab, a humanized immunoglobulin G monoclonal antibody that binds to IL-5 (Figure 3b), was generally safe and well tolerated in a study conducted by Kang et al. in their phase 2 multicenter, randomized, double-blind, placebo-controlled, parallel-group clinical trial investigating the efficacy and safety of subcutaneous injection of mepolizumab 100 mg every 4 weeks for 16 weeks. The primary endpoint of the study was the proportion of patients who achieved an IGA of 0 or 1 and at least a 2-grade improvement at week 16. In their study, only 2 out of 18 patients enrolled in the mepolizumab group achieved an IGA of 0 or 1. While a greater proportion of patients achieved ≥50% and ≥75% improvements in score from baseline at week 4 and week 8, this was not statistically significant. This prompted the group to conclude that mepolizumab did not show clinical improvements in patients with moderate-to-severe AD. Mepolizumab may be effective in preventing flares rather than controlling an active disease [86]. While no safety concerns were noted in a proof-of-concept study, mepolizumab failed to reach the pre-determined futility criteria after interim analysis [87].

The investigation of benralizumab, which directly binds to the IL-5Rα chain (Figure 3b) to inhibit the signaling pathway, was recently conducted in a phase 2 multinational study. This study is a randomized, double-blind, parallel-group design that included a 16-week placebo-controlled phase, followed by a 36-week extension phase. A total of 194 patients with moderate-to-severe AD, despite prior treatment with topical medications, were enrolled in the study (referred to as the HILLIER study). The primary endpoint of the study was the proportion of patients who achieved an IGA score of 0/1 and a decrease in IGA of ≥2 points at week 16 compared to baseline. However, the study was terminated [88].

#### 2.3.3. Interleukin-22 (IL-22)

IL-22 cytokine is produced by mast cells, various in vitro and animal studies propose its role in epidermal hyperplasia, and it contributes to skin barrier damage by the inhibition of keratinocyte differentiation. This cytokine is also responsible for the upregulation of pruritogenic peptide gastrin-releasing peptide (GRP), TSLP, and interleukin 33 in epithelial cells, as shown in Figure 1a and Figure 3a [89].

A phase 2a randomized double-blind trial was conducted to evaluate the effectiveness of fezakinumab, an anti-IL-22 monoclonal antibody (Figure 3b). The trial involved intravenous fezakinumab monotherapy administered every 2 weeks for a duration of 10 weeks, with follow-up assessments conducted until 20 weeks. The study findings revealed a statistically significant decline in SCORAD of 21.6 ± 3.8 in fezakinumab-treated patients compared to 9.6 ± 4.2 in placebo-treated patients at 12 weeks. Similarly, there was a statistically significant decline in IGA of 0.7 ± 0.2 in fezakinumab-treated patients compared to 0.3 ± 0.1 in placebo-treated patients at 12 weeks. These results demonstrate improvements in SCORAD and IGA scores among patients with severe AD but not in those with moderate AD [90]. While the trial looked promising, it only has a limited sample size. More studies should be conducted in targeting this particular interleukin.

#### 2.3.4. Phosphodiesterase-4 (PDE4)

The phosphodiesterase (PDE) superfamily is composed of 11 isozymes that degrade cyclin nucleotides, particularly cAMP and cGMP (Figure 3a) [91]. PDE4 is the most diversified and is expressed in keratinocytes, T-cells, Langerhans cells, and neutrophils. PDE4 includes four subtypes, PDE4A, PDE4B, PDE4C, and PDE4D, of which PDE4B plays a significant role in inflammatory response [92]. Inhibition of PDE4 results in the prevention of degradation of intracellular cAMP, thereby activating protein kinase A (PKA), cyclic nucleotide-gated ion channels, and the exchange factor directly activated by cAMP 1 and 2 (Epac1/Epac2) at the same time, promoting the production of anti-inflammatory cytokines through interactions with the cAMP-responsive element binding protein (CREB) [93]. Among the immune functions are the following: regulation of pro-inflammatory and anti-inflammatory cytokines, activation of T cells, neutrophil degranulation, performance of antigen-presentation, epithelial integrity via initiation of multiple downstream elements, and suppression of the production of reactive oxygen species [93]. Hence, targeting this has been beneficial for inflammatory conditions like AD and psoriasis.

##### Topical Phosphodiesterase-4 (PDE4)

Crisaborole, a topical phosphodiesterase-4 inhibitor (Figure 3b), has been approved by the FDA for the treatment of mild to moderate AD in pediatric and adult patients. The safety of crisaborole 2% ointment has been established for up to 1 year of treatment, with skin burning being the most commonly reported adverse effect.

Several selective phosphodiesterase 4 inhibitors are currently under development. One of these investigational compounds is PF-07038124 ointment, a topical small-molecule PDE4 inhibitor. Currently, a phase 2b randomized, double-blind, vehicle-controlled, parallel-group study is underway to assess the efficacy, safety, tolerability, and pharmacokinetics of multiple dose levels (0.01%, 0.03%, and 0.06%) of PF-07038124 ointment, an oxaborole-based PDE-4 inhibitor (Figure 3b). The study aims to recruit approximately 246 participants aged 12 years and older with mild-to-moderate AD or mild-to-severe plaque psoriasis. Participants will apply PF-07038124 0.01% ointment once a day for a duration of 12 weeks [94].

In a phase 2, multicenter, randomized, blinded, excipient, parallel-group clinical study, topical small-molecule PDE4 inhibitor Hemay-808 ointment (1%, 3% or 7%) (Figure 3b) or a vehicle was to be applied twice daily for 28 days in 148 adult participants (18 to 65 years old) diagnosed with mild to moderate AD. The study is still in the recruitment phase with no available results as of this writing [95].

Difamilast is a selective PDE4 inhibitor (Figure 3b) in topical form and was approved for the treatment of AD in Japan in 2021. Two phase 3 trials have been recently completed in Japan for both adult and pediatric populations. In a phase III randomized, double-blind, vehicle-controlled trial by Sakai et al., patients aged 2–14 years with an IGA score of 2 or 3 received difamilast 0.3% (n = 83), difamilast 1% (n = 85), or a vehicle (n = 83) ointment twice daily for 4 weeks. The success rates in IGA score, which is the primary endpoint in their study at week 4, were 44.6%, 47.1%, and 18.1% in the difamilast 0.3%, difamilast 1%, and vehicle groups, respectively. The improvement in EASI, which is their secondary endpoint, was significantly higher in patients in the difamilast 0.3% and 1% group as compared with the vehicle group, prompting the researchers to conclude that in the pediatric study, the IGA response at 4 weeks was higher in both difamilast groups, and EASI-75 achievement is also significantly higher in active arms. Safety profiles in pediatric patients were similar with those in adult patients. Currently, a phase III, multicenter, open-label, uncontrolled study in infants (<2 years) is ongoing [96]. Saeki et al., in their phase 3 randomized, double-blind, vehicle-controlled trial, evaluated patients aged 15–70 years old with AD with an IGA of 2 or 3 to receiving either topical difamilast ointment 1% (n=182) or vehicle (n=182) twice daily for 4 weeks. The percentage of patients achieving IGA scores of 0 or 1 was significantly higher in adult patients on difamilast therapy (38.46%) versus the placebo group (12.64%). EASI-75 was obtained in about 42.86% of difamilast-treated patients as compared with 13.19% in the placebo group. The most frequently encountered TEAE was the worsening of AD (3.8% in difamilast-treated patients versus 12.1% in vehicle) and nasopharyngitis (4.9% in difamilast-treated patients versus 3.8% in vehicle) [97].

Roflumilast is a highly potent PDE-4 inhibitor (Figure 3b) and its 0.3% cream form was approved by the US FDA for the treatment of plaque psoriasis in adolescents and adults [98]. A 4-week, phase II proof-of-concept study that involved 136 AD patients ≥12 years was randomized to a once daily application of roflumilast cream 0.15%, roflumilast cream 0.05%, or placebo for 4 weeks. The primary endpoint, represented by the absolute change from baseline in EASI score at week 4, was not achieved in their study (−6.4 in roflumilast 0.15%, −6.0 in roflumilast 0.05%, and −4.8 in vehicle) [99]. Two parallel-group, double-blind, vehicle-controlled phase 3 trials (Integument-I and Integument-II) in which roflumilast 0.15% cream versus a vehicle was applied once daily for 4 weeks in subjects with AD were started in February 2021 and were completed in 2022 [100]. Although the phase 2 study did not reach its primary endpoint, that is, a significant reduction in EASI score, two parallel phase 3 trials were completed. However, no published results are available as of this writing. A phase 3, 4-week, parallel-group, double-blind, vehicle-controlled study of the safety and efficacy of roflumilast cream 0.05% administered once a day in subjects ages 2–5 years old with AD (Integument-PED) is ongoing [101].

##### Oral Phosphodiesterase-4 (PDE4)

Simpson et al. in their phase II, double-blind, placebo-controlled trial evaluated the efficacy, safety, and pharmacodynamics of apremilast in adults with moderate to severe AD. In their study, patients were randomly assigned or received placebo, apremilast 30 mg twice daily, and apremilast 40 mg twice daily for 12 weeks. During weeks 12 to week 24, all patients would receive either apremilast 30 mg or apremilast 40 mg. Among the 185 randomly assigned intention-to-treat patients at week 12, apremilast 40 mg revealed a statistically significant improvement in EASI (31.6%) as compared with placebo and apremilast 30 mg. Furthermore, apremilast 40 mg was associated with a more unfavorable adverse-event profile, including nausea, diarrhea, headache, and cellulitis, leading to the termination of the study [102].

A phase 2b dose-ranging study is currently underway, examining the effects of orismilast-modified release tablets in adults (18 years and older) with moderate to severe AD. The study follows a randomized, double-blind, placebo-controlled, parallel-group design, involving 210 participants. During the 16-week treatment period, participants are administered with orismilast-modified release tablets at varying doses (20 mg, 30 mg, or 40 mg) or placebo, twice daily (morning and evening). The study is currently in the recruitment phase [103]. The mechanism of action of these drugs is depicted in Figure 3b.

#### 2.3.5. Histamine Receptors

Histamines are well-known pruritogens and causes the release of mast cells and basophils via the activation of histamine receptors. There exist four subtypes of antihistamines, those of particular importance of which are H1 and H4 antihistamines, which play roles in pruritic skin diseases like AD and urticaria. Human CD4(+) T cells express a functional H4 receptor, which was upregulated under Th2 conditions in a study by Gutzmer et al. [104]. Numerous studies demonstrated the anti-pruritic effects of H1R antihistamines in urticaria, and their efficiency in AD is still limited. H4 receptors (H4Rs), the most recently discovered subtype, are expressed on keratinocytes (Figure 3a), neurons (Figure 4a), and various immune cell populations like peripheral mononuclear leukocytes and exert immuno-regulatory effects through the upregulation of IL-31, a T-cell-derived cytokine strongly linked with the development of skin barrier dysfunction, local inflammation, and pruritus [105]. Emerging clinical studies specifically targeted the H4R subtype to control the pruritus. In vivo studies demonstrated reduced Th2 cytokine production, associated pruritus, and skin inflammation by H4R antagonists in AD-associated animal models.

The efficacy and safety of JNJ 39758979, a potent and selective H4R antagonist (Figure 3b and Figure 4b), were assessed by Kollmeier et al. in a study involving healthy subjects with histamine-induced pruritus. The researchers concluded that for the treatment of pruritus that is not adequately controlled by H1R antihistamines, novel H4R antagonists like JNJ 39758979 offer a viable alternative treatment option [106]. In another study, Kollmeier et al. was able to demonstrate the potential benefit of JNJ-39758979 300 mg on lung function and asthma control in eosinophilic asthma patients [107].

Adriforant is a functional antagonist of histamine receptor 4 and is also known as ZPL 3893787, PF 3893787, and ZPL 389. ZPL-3893787 30 mg, an oral selective H4R antagonist (Figure 3b and Figure 4b) given once a day for 8 weeks, has been shown to exert anti-inflammatory effects over placebo in patients with physician-documented moderate-to-severe AD [108]. Subsequently, a multicenter double-blinded parallel trial comparing the safety and efficacy of adriforant (ZPL-3893787) 30 mg once a day and 50 mg once a day versus placebo in patients with concomitant use of topical corticosteroids or topical calcineurin inhibitor in patients with moderate to severe AD was not able to fulfill the pre-specified efficacy criteria and was eventually terminated [109].

A potent and selective H4R antagonist, LEO 152020 tablet (Figure 3b and Figure 4b), is currently being tested in a phase 2 clinical trial compared with placebo tablets for up to 16 weeks in the treatment of 224 adult participants with moderate to severe AD. The study aims to provide valuable information regarding the validity of the H4 receptor as a potential therapeutic target in AD [110].

While the pathophysiology of pruritus is not limited to H4 receptors, a combined treatment targeting H1 and H4 receptors might be a good strategy in treating patients with AD [111].

#### 2.3.6. Molecules Involved in Migration of T-Cells

##### Circulating Memory CLA+ T Lymphocytes

CLA+ T cells are produced during the initiation of the AD lesion. These cells require chemo-attractant stimuli to function through their chemokine receptors CCR4 and CCR10. Thymus and activation-regulated chemokine (TARC/CCL17) and macrophage-derived chemokine (MDC/CCL22) are the chemokine ligands specific for CCR4 (Figure 1a), while the cutaneous T-cell-attracting chemokine (CTACK/CCL27) is specific for CCR10 [112]. Some chemokines may be responsible for the selective recruitment of Th2 cells in the Th2-predominant inflammatory diseases like AD, as demonstrated in Figure 1. The expression of these chemokines and chemokine receptors correlates well with the clinical severity of AD; hence, the inhibition of this migration might be a potential therapy in treating AD [113].

RPT 193 is a CCR4 antagonist (Figure 1b) and a drug candidate administered orally. Reports from a first-in-human, three-part, multi-center, phase 1, randomized, double-blind, placebo-controlled study in 64 healthy male and female subjects and 30 male and female patients with AD showed that it was safe and well tolerated but showed mild adverse effects after single and multiple dosing of RPT193 [114]. Currently, a randomized, placebo-controlled phase 2 study of RPT193 (50 mg, 200 mg, 400 mg) oral tablets versus placebo administered daily for 16 weeks in 268 adults (18 years to 75 years) with moderate-to-severe AD is recruiting participants and is estimated to be completed by September 2023 [115].

##### Sphingosine-1-Phosphate (S1P)

Pro-inflammatory lipid sphingosine 1-phosphate (S1P) is a bioactive lipid mediator released from red blood cells and acts as a second messenger that is involved in several immunological processes like angiogenesis, cytoskeleton organization, trafficking of immune cells, mitogenesis, and apoptosis through acting on either one of the five sphingosine-1-phosphate-specific G-protein coupled receptors (S1PRs) (Figure 1a, Figure 3a and Figure 4a): sphingosine-1-phosphate receptor 1 (S1PR1), sphingosine-1-phosphate receptor 2 (S1PR2), sphingosine-1-phosphate receptor 3 (S1PR3), sphingosine-1-phosphate receptor 4 (S1PR4), and sphingosine-1-phosphate receptor 5 (S1PR5). S1PR1, S1PR2, S1PR3, S1PR4, and S1PR5 are expressed in keratinocytes, while Langerhans cells express S1PR1, S1PR2, and S1PR4, as reflected in Figure 1. S1P plays a diverse role in different cell types, but S1PR1 is expressed by most immune cells [116]. Although S1P has been associated with pain and hypersensitivity, the role of S1P in pruritus has not been well established. In a mouse study by Hill et al., administering high levels (2–10 µm) of S1P induced scratching and pain-like behavior, while low levels (0.2 µm) induced only scratching. The group was able to demonstrate the role of S1PR3 in the activation of itch and pain neurons and the importance of S1P/S1PR3 [117]. Sphingosine, a molecule that exerts an antimicrobial effect, is found to be decreased in patients with AD, hence making them susceptible to staphylococcus aureus colonization [118]. The reduction in ceramide level contributes to the dysregulation of sphingolipid metabolism in conditions like AD, psoriasis, and ichthyosis, leading to disease progression and severity [117]. Patients with AD have decreased larger ceramides and elevated smaller ceramides, contributing to an impaired skin barrier [119]. Sakai and Bieber evaluated serum levels of S1P from 17 patients with AD, 22 non-AD patients with atopic comorbidities (allergic rhinitis, bronchial asthma, and/or food allergy), and 47 healthy controls. They were able to demonstrate elevated serum S1P in AD patients with EASI ≥16 [119].

Topical S1PR antagonist

KRO-105714 (Figure 1b, Figure 3b and Figure 4b), a dual antagonist of sphingosylphosphorylcholine and S1PR1 in topical form, was able to show showcase its safety and efficacy in multiple in vivo AD models and alleviate its symptoms through reduction in pro-inflammatory cytokines and chemokines [120].

The topical application of JTE-013, an S1PR2 antagonist (Figure 3b and Figure 4b), was able to significantly reduce inflammation and mast cell accumulation in skin tissues of 2,4-dinitrochlorobenzene (DNCB)-induced AD mouse models, suggesting it as a potential therapeutic agent in AD patients [121].

2.Oral S1PR modulator

In a phase 2, randomized, double-blind, placebo-controlled, parallel-group study investigating 140 adult (18 years and older) participants with moderate to severe AD, etrasimod (Figure 1b, Figure 3b and Figure 4b) and oral S1PR modulator tablets (2 mg or 1 mg) or placebo were given once a day for 12 weeks. The primary outcome was the percent change in EASI score from baseline. Although the primary outcome was not met, efficacy was demonstrated by estrasimod 2 mg by several clinician- and patient-assessed measurements, and both 1 mg and 2 mg were well tolerated [122].

LC51-0255 (LG Chem) is a potent, selective, and orally available S1P1 receptor modulator (Figure 1b, Figure 3b and Figure 4b) that reduces peripheral absolute lymphocyte count (ALC) by preventing the recirculation of lymphocytes from lymphatic tissue to the target organs. A randomized, double-blind, placebo-controlled, dose-escalation study was conducted in 50 healthy subjects wherein each subject received LC51-0255 (0.25, 0.5, 1, 2, or 4 mg) orally or its matching placebo in an 8:2 ratio. Their study was able to conclude that LC51-0255 has a long half-life, is well tolerated, and reduces ALC in a dose-dependent and reversible manner. These findings are a beneficial treatment option for patients with autoimmune disease [123].

Udifitimod (BMS-98166) acts by targeting S1PR1 (Figure 1b, Figure 3b and Figure 4b). In August 2021, a phase 2, randomized, double-blinded, placebo-controlled, 5-parallel-group study evaluated the efficacy, safety, and tolerability of BMS-986166 or branebrutinib for the treatment of patients with moderate to severe AD evaluated, but its result has not been posted yet [124]. In a recent article, GlobalData announced that udifitimod (BMS-986166), which is currently in phase II and is a drug for Atopic Dermatitis (Atopic Eczema), has a 31% phase transition success rate (PTSR) indication benchmark for progressing into Phase III [125].

SCD-044, a novel orally bioavailable S1PR1 agonist (Figure 1b, Figure 3b and Figure 4b), is currently being studied due to its effect of inducing a diminished sequestration of lymphocytes out of the lymphatic tissue, therefore reducing inflammation. A phase 2 randomized, double-blind, placebo-controlled study is currently investigating the safety and efficacy of SCD-044 tablets (three different doses: low dose, intermediate dose, and high dose) for 16 weeks in 240 participants with moderate-to-severe AD [126].

Numerous in vivo and in vitro studies have been suggesting the role of the S1P pathway in AD, making it a potential therapeutic agent in humans.

#### 2.3.7. Bruton’s Tyrosine Kinase (BTK)

Expressed in all hematopoietic cells except T cells, Bruton’s tyrosine kinase (BTK), a member of the TEC kinase family of non-receptor tyrosine kinases, has been a promising target for immunological disorders like AD and psoriasis, as depicted in Figure 3a [127]. BTK is a multi-component signaling protein that not only aids in the differentiation of B-cells but contributes with innate and adaptive immunity and, more importantly, cytokine production. BTK inhibitors (BTKis) suppress B-cell receptor- and myeloid-fragment-crystallizable-receptor-mediated signaling, thus inhibiting B-cell activation, antibody class-switching, expansion, and cytokine production [127].

In a study by Xing et al., topical PRN473, a BTK inhibitor (Figure 3b), effectively suppressed IgG- and IGE-mediated signaling, hence preventing the downstream immune effects and recruitment of macrophages and mast cells [128]. PRN473, a covalent BTK inhibitor in topical form applied in multiple topical doses for 42 days, was evaluated versus placebo in 39 adult (18 to 70 years of age) patients with mild to moderate AD in a Phase 2a randomized, intra-patient, double-blind, placebo-controlled study. No results as of this writing are available yet, but recruitment has already been completed [129].

Branebrutinib is an oral and highly selective BTK inhibitor (Figure 3b) that covalently binds to the cysteine residue of BTK [130]. A phase 2, randomized, double-blind, placebo-controlled, 5-parallel-group study trial was designed to evaluate the efficacy, safety, and tolerability of oral branebrutinib for 16 weeks in 17 adult (18 to 65) participants with moderate to severe AD. No results are available yet [124].

A phase 2, randomized, double-blind, placebo-controlled, multicenter proof-of-concept study on rilzabrutinib, a BTK inhibitor (Figure 3b), is currently recruiting 136 adult (18 years and older) patients with moderate-to-severe AD who are inadequate responders or intolerant to topical corticosteroids. Rilzabrutinib twice a day or three times a day will be given for 16 weeks and the estimated study completion date is November 2024 [131]. Targeting a significant pathogenic pathway like the BTK pathway rather than a single molecule represents a novel therapeutic approach in patients with autoimmune dermatological conditions and may have the potential to address previously unmet needs [132].

#### 2.3.8. Liver X Receptor (LXR)

Liver x receptor (LXR), a member of the nuclear receptor superfamily, is expressed in keratinocytes (Figure 3a) and fibroblasts and exists in alpha and beta isoforms. LXR forms heterodimers with RXR, enabling it to regulate various functions such as maintaining epidermal homeostasis and reducing inflammatory responses [133]. Being able to exert these functions makes these receptors potential targets for pharmacological intervention in AD.

In a phase II study with a double-blind, placebo-controlled design, involving 104 patients with mild-to-moderate AD, the topical LXR-β agonist VTP-38543 (Figure 3b) was administered twice daily for 28 days at varying concentrations (0.05%, 0.15%, and 1.0%). Although the drug successfully enhanced the mRNA expression of structural proteins such as loricrin and filaggrin, and led to a reduction in epidermal thickness, it did not demonstrate the ability to downregulate the Th2/Th17 markers [134].

In a multicenter, randomized, double-blind, bilateral, vehicle-controlled study investigating 209 adult and adolescent subjects with moderate AD, ALX-101 (Figure 3b) at different concentrations (1.5% and 5%) was applied for 42 days. While the study has been completed, the results are not yet available [135]. Currently, another phase 2, randomized, double-blind, vehicle-controlled, parallel-group study to evaluate the safety and efficacy of ALX-101 gel 5% and a matching ALX-101 gel vehicle applied topically twice daily for 56 days in 124 adult and adolescent (12 years and older) subjects with moderate AD is active but not recruiting. The results are still not available [136].

### 2.4. Targeting the Itch–Scratch Cycle

The pyschosomatic aspect of the itch–scratch cycle greatly affects the quality of life of patients with AD. This cycle disrupts the epidermal barrier, causes damage to keratinocytes, activates local dendritic cells, and triggers the adaptive response.

#### 2.4.1. Interleukin-31 (IL-31)

Coined as the “itchy” cytokine, IL-31 has been associated with acute itch in an article by Oetjen et al. [68] and has been an important mediator of pruritus in chronic conditions like AD and prurigo nodularis [137]. Once pro-inflammatory IL-31 cytokine binds to either of the heterodimer receptors IL-31 receptor A (IL31RA) and oncostatin M receptor beta subunit (OSMRβ), it exerts its role in inflammation, pruritus, immune defense, and tissue hemostasis, as showcased in Figure 3a [138]. IL-31RA is expressed in epithelial and neuronal cell types, while OSMRβ is widely expressed throughout the mammalian body [138]. Intradermal injection of IL-31 induced epidermal thickness in murine studies. The study was able to demonstrate the crucial role of IL-31 in cell proliferation, inflammation, and maintaining skin integrity. The overexpression of IL-31 increased epidermal thickness and transepidermal water loss in the skin [139].

In a post hoc analysis of a Phase III randomized controlled trial conducted in Japan, nemolizumab, a humanized monoclonal antibody that antagonizes IL-31RA (Figure 3b), was evaluated in two parts (Part A and Part B). Part A involved the subcutaneous administration of nemolizumab at a dose of 60 mg every 4 weeks for 16 weeks, followed by a long-term follow-up period of up to 68 weeks (Part B). The study included patients aged 13 years and above with AD. The primary endpoint of Part A was the mean percent change from baseline in the pruritus visual analog scale (VAS) score at week 16. The analysis revealed a reduction of 42.8% in the nemolizumab group receiving topical agents, compared to 21.4% in the placebo group receiving topical agents. Additionally, the percentage change in the EASI score from baseline to week 16 was 45.9% in the nemolizumab group with topical agents, and 33.2% in the placebo group with topical agents. These results demonstrated that nemolizumab led to a greater reduction in pruritus compared to placebo, achieving the primary endpoint of the study [140]. Kabashima et al. conducted two phase III long-term studies, namely Study-JP01 and Study-JP02, to evaluate the efficacy of nemolizumab 60 mg administered subcutaneously every 4 weeks in patients with AD and moderate-to-severe pruritus. Study-JP01 was a 52-week open-label long-term extension period study, which served as Part B of the study. In this phase, all patients received nemolizumab 60 mg every 4 weeks until week 64. The treatment groups included nemolizumab for 16 weeks followed by nemolizumab for 52 weeks, and placebo for 16 weeks followed by nemolizumab for 52 weeks. Study-JP02 involved patients who received nemolizumab 60 mg every 4 weeks up to week 52. The analysis of Study-JP01 demonstrated a significant reduction in pruritus visual analog scale (VAS) by 66% throughout the 68-week treatment period, starting from the initiation of treatment. This reduction was statistically significant compared to the previous study, which reported a reduction of 42.8%. Furthermore, the group observed a decrease in EASI score by 78% from the group that received nemolizumab for 16 weeks followed by nemolizumab for 52 weeks, extending up to week 68 [141].

A Phase IIa randomized, double-blind, placebo-controlled study by Sofen et al. investigated the efficacy and safety of vixarelimab, a first-in-class fully human monoclonal antibody that targets the oncostatin M receptor (Figure 3b) in 49 participants with moderate-to-severe prurigo nodularis. Vixarelimab 360 mg given subcutaneously (loading dose of 720 mg at week 0) was administered weekly for 8 weeks versus placebo. The results of their study were promising as it provided a rapid, statistically significant, and clinically meaningful reduction in pruritus in patients with prurigo nodularis. Clear to almost clear skin was achieved at 8 weeks in approximately one-third of patients. In their study, the oncostatin M receptor pathway was believed to be responsible for the pathogenesis of fibrosis in prurigo nodularis. The groups also concluded that the improvement in the Worst Itch–Numeric Rating Scale (WI-NRS) score was possible due to vixarelimab blocking the shared oncostatin M receptor beta subunit between IL-31 and oncostatin M receptor pathway [142].

#### 2.4.2. Neurokinin 1 Receptor (NK1R)

NK1Rs are channels located on mast cells and the dorsal horn of the spinal cord, as pictured in Figure 1a and Figure 4a [143]. Once activated by substance P, it leads to the sensitization of mast cells, which leads to increased expression of TNFα, which sensitizes type C nociceptors [144]. The blocking of substance P may be beneficial to interfering with the cross-talk between the mast cells and the nerves, which is responsible for the pruritus [145].

A phase 3 randomized, placebo-controlled, double-blind clinical trial, known as the EPIONE study, investigated the efficacy of oral tradipitant (VLY-686), a novel NK-1 receptor antagonist (Figure 4b), in reducing chronic pruritus in adults with mild to moderate AD. Although the study did not achieve its primary endpoint of pruritus reduction, statistically significant improvements were observed in itch after one day of treatment and in sleep after two days of treatment in the population with mild AD. The findings suggest the need for future studies to refine treatment recommendations for patients with mild lesions who experience significant pruritus [146].

In a large Phase II study called the ATOMIK study, involving 484 participants with pruritus caused by AD, the oral NK1 antagonist selropitant (Figure 4b) was administered at doses of 1 mg or 5 mg. The treatment regimen included a loading dose of three tablets at bedtime, followed by a daily tablet at bedtime for 6 weeks. However, the study did not achieve its primary endpoint, which was the change in the Worst Itch–Numeric Rating Scale (NRS) from baseline [147].

The inhibition of sensory-neuron-associated neurokinin-1 receptors presents a promising target for drug therapy as the elevated expression of these receptors has been linked to the development of pruritus in AD.

#### 2.4.3. P2X Purinoreceptor3 (P2XR3)

Extracellular adenosine 5′-triphosphate (ATP) is released from cells under physiologic or pathologic conditions. These ATPs are mediated by cell-surface receptors called P2R, which are divided into two families: P2XR and P2YR. P2XR has several heterotrimers, among which P2XR3 is particularly important in this review article, which are cationic channels mainly expressed in sensory neurons, as illustrated in Figure 4b. These receptors play a role by the localized release of pro-inflammatory neuropeptides via the axon reflex, resulting in coughing, peripheral irritation, pain sensation [148], and possibly itch.

In an AD mice model, intradermal treatment with A317491 (selective P2XR3 antagonist) (Figure 4b) reduced spontaneous scratching by approximately 30%, demonstrating the role of targeting the ATP-P2XR3 signaling pathway in improving chronic itch [149].

In the BLUEPRINT trial, a Phase II multicenter study, the efficacy of BLU-5937—a selective P2XR3 antagonist (Figure 4b)—was evaluated in the treatment of chronic pruritus in 142 adult participants with AD. The study followed a randomized, double-blind, placebo-controlled, parallel design, where participants were given BLU-5937 at a dose of 200 mg twice a day for 4 weeks, while others received a placebo. Although BLU-5937 was found to be safe and well tolerated, it did not demonstrate statistical significance for its primary endpoint, which was the reduction in the weekly mean Worst Itch–Numeric Rating Scale (WI-NRS) [150].

#### 2.4.4. Transient Receptor Potential Channel (TRP)

The TRP channel family is divided into eight subfamilies, including TRPA (ankyrin), TRPC (canonical), TRPM (melastatin), TRPML (mucolipin), TRPN (Drosophila NOMPC), TRPP (polycystin), TRPV (vanilloid), and TRPY (yeast) according to their amino acid sequence. These receptors are involved in transmitting sensory inputs such as heat, pain, and taste. These channels are highly expressed in the keratinocytes, mast cells, cutaneous sensory neurons, and T-cells, as showcased in Figure 3b and Figure 4b [151]. TRPA and TRPV channels hold significance in the context of AD.

##### Transient Receptor Potential Channel Ankyrin (TRPA)

Transient receptor cation ankyrin 1 (TRPA1) is a novel channel that is believed to be responsible for the histamine-independent itch pathway in AD. Aside from the TRPA1 receptor being a cold-sensitive calcium channel in keratinocytes in response to temperatures less than 17 °C, these channels are also activated by various endogenous (leukotriene B4, TSLP, serotonin, IL-13, and IL-31) and exogenous pruritogens (chloroquine, cowhage, allyl isothiocyanate, cinnamaldehyde, allicin, and carvacrol). Once activated, it triggers the different pruritogenic pathways, as seen in Figure 3a and Figure 4a [152]. Oh et al. conducted a study and discovered that TRPA1 channels exhibit high expression in dermal cells, sensory afferents, and keratinocytes, and are co-expressed with tryptase+ mast cells in lesional skin biopsies of patients with AD following IL-13-induced itch. These findings suggest the potential of TRPA1 channels in future therapeutic approaches for chronic itch [153].

Topical TRPA1 antagonist HC-03003 (Figure 3b and Figure 4b), applied after irradiation, blocked the development of mechanical and thermal allodynia in a murine study by Fialho et al. [154].

Another TRPA1 antagonist, oral GRC17536 (Figure 3b and Figure 4b), was able to finish a phase II clinical trial as a promising treatment for diabetic neuropathy but was not able to advance to phase III, due to its limited bioavailability and pharmacokinetic profile [155].

##### Transient Receptor Potential Channel Vanilloid (TRPV)

TRPV channels have six members, TRPV1, TRPV2, TRPV3, TRPV4, TRPV5, and TRPV6. Vanillin, vanillic acid, and capsaicin are among the substances that trigger these channels [156]. TRPV 1–4 are non-selective cation channels sensitive to temperature, while TRPVs 5–6 are highly calcium-selective channels not sensitive to temperature [157].

Transient receptor potential channel vanilloid 1 (TRPV1)

TRP cation channel subfamily V member 1 (TRPV1) is one of the extensively studied channels. TRPV1 and TRPA1 are both expressed on primary afferent sensory neuron (Figure 4a). When TRPV1 is activated, it leads to the release of substance P and calcitonin-gene-related peptide, resulting in an increased itch sensation. Additionally, these channels are expressed on cutaneous sensory neurons and dorsal root ganglions, and other non-neuronal cells, such as keratinocytes, monocytes, macrophages, mast cells, neutrophils, and T-cell, and they play a role in mediating pruritus induced by interleukin-31. It is worth mentioning that dendritic cells and eosinophils only have TPRV1 [158].

Several clinical trials have provided evidence of the safety of topical agents targeting TRPV1. Topical PAC-14028, a TRPV1 channel inhibitor (Figure 4b), has been investigated in a phase IIb randomized, double-blind, multicenter, placebo-controlled trial conducted by Lee et al. The study involved 194 adult patients with mild to moderate AD who applied PAC-14028 at concentrations of 0.1%, 0.3%, or 1.0% twice daily for 8 weeks. The results showed that PAC-14028 exhibited superiority over the vehicle, with statistically significant improvements observed in various measures, including IGA success rates, SCORAD index, EASI 75/90, sleep disturbance score, and pruritus visual analog scale [159].

Another drug is CT327/SNA-120, a topical TRPV1 pathway inhibitor (Figure 4b) that demonstrated its safety and efficacy in a phase IIb clinical trial for psoriasis [160].

The CAPTAIN-AD study, a phase 3 trial, aimed to evaluate the potential of asivatrep, a topical TRPV1 antagonist (Figure 4b), as an alternative treatment for adolescents and adults with AD. While topical asivatrep applied over the 8-week, twice-a-day treatment in 240 patients with adolescent (aged 12 years and older) and adult AD was well tolerated and significantly improved the mean percentage change of EASI from baseline compared with the vehicle, the study still hopes to address the efficacy and safety of long-term treatment. Furthermore, topical asivatrep, a potent and selective TRPV1 antagonist, did not induce hyperthermia during the 8-week, twice a day treatment [161].

By targeting this channel, the release of pruritic and pro-inflammatory mediators can be suppressed, leading to the restoration of the skin barrier.

2.Transient receptor potential channel vanilloid 3 (TRPV3)

In the study of Seo et al., the group showed that keratinocytes isolated from patients with AD exhibited enhanced expression and heat sensitivity with a hyperactive channel function of TRPV3 (Figure 4a). This stimulation leads to increased scratching behavior and produced higher levels of thymic stromal lymphopoietin, nerve growth factor, prostaglandin E_2_, and IL-33 from the epidermis, which were attenuated by pharmacologic inhibition of TRPV3 [162].

While TRPV1 has been extensively studied, the TRPV3 channel is now gaining recognition as a promising focus for skin regeneration. This is attributed to its impact on epidermal regeneration through EGFR signaling, specifically via the phosphatidylinositol 3 kinase (PI3K)/protein kinase B (AKT) pathway. Numerous investigations are being conducted to explore the potential of TRPV3 in this context [163].

#### 2.4.5. Cannabinoid Receptors (CBRs)

The endocannabinoid system (ECS) is an evolutionary complex intercellular signaling network, which plays roles in the modulation of the immune and nervous systems. The two main receptors for endocannabinoids (ECB) are the cannabinoid type 1 (CB1R) and cannabinoid type 2 (CB2R) receptors, which are G-coupled proteins expressed in epidermal keratinocytes, melanocytes, dermal cells, mast cells, sweat glands, hair follicles, and cutaneous nerve fibers, as shown in Figure 3a and Figure 4a [164,165]. In the epidermal keratinocytes, the CB1R located in the stratum granulosum and stratum spinosum, once activated, triggers cytokine storms, which intensifies the generation of reactive oxygen species (ROS) and TNFα. On the other hand, the CB2R located in the basal layer of epidermal keratinocytes, once activated, intensifies the body’s anti-inflammatory response via reduction in the generation of ROS and TNFα. Hence, targeting CB2R may prove to be beneficial in the treatment of AD [166]. Improved epidermal barrier function, decreased Th2 inflammatory response, and suppressed mast cell production were observed after activating CB1R in various experimental mouse models of AD [167,168,169]. Several studies and clinical trials have shown that cannabinoids and cannabinoid-containing oils have been helpful in alleviating the symptoms of AD such as pruritus, irritation, and skin dryness.

A Phase Ib/IIa trial, employing a multiple-dose, double-blind, randomized placebo-controlled design, was conducted to assess the efficacy of oral CBR2 agonist S-777469 (Figure 3b and Figure 4b) in individuals with mild to moderate AD. The trial involved 37 adult participants aged 18 to 65 years who received either S-777469 (at doses of 50 mg, 200 mg, or 800 mg) or a vehicle twice a day for a duration of 14 days. Despite the initial promise associated with targeting the receptor, the findings of this study demonstrated the ineffectiveness of the drug in treating AD [170].

Due to the similar ligand-binding between CBR1 and CBR2, targeting the CBR2 ligands remains to be a therapeutic challenge. Given the neuro-skin physiology pathways as one of the multifactorial factors of AD, future studies developing topical cannabinoids with greater polarity and higher CBR2 selectivity must be taken into consideration [171].

#### 2.4.6. Protease-Activated Receptor 2 (PAR2)

The family of protease-activated receptors (PARs) includes protease-activated receptor-1 (PAR-1), protease-activated receptor-2 (PAR-2), protease-activated receptor-3 (PAR-3), and protease-activated receptor-4 (PAR-4). In vivo and in vitro studies suggest the roles of PARs in the regulation of epidermal permeability and barrier function [172].

Protease-activated receptor-2 (PAR-2), a G-protein coupled receptor expressed in keratinocytes, neurons (Figure 4a), and inflammatory cells such as mast cells (Figure 1a) and T-cells, has been demonstrated to be of significance in AD [173]. PAR-2 overexpression in murine studies exhibited an increased density of nerve fibers [174] and skin remodeling [175]. During skin inflammation, proteolytic enzymes trypsin and tryptase signal pro-inflammatory factors through PAR-2, leading to the production of chemokines and cytokines like TNFα, IL-4, and TSLP [173,176]. In addition, the activation of PAR-2 on keratinocytes and endothelia stimulates the NF-κb signaling pathway, which has been suspected to be linked with AD. In the study of Steinhoff et al., PAR-2 was significantly enhanced on skin biopsies of 38 patients with AD. On the other hand, tryptase (endogenous PAR-2 agonist) was increased up to fourfold [177]. Nishimoto et al. also demonstrated the crosstalk between PAR-2- and IL-4-mediated inflammatory axes, hence being a potential platform for mechanism-targeted drugs [176]. Furthermore, proteases may also activate downstream signaling in the keratinocytes, promoting the release of IL-8 [178].

Topical methylbenzyl methylbenzimidazole piperidinyl methanone (MMP), a selective PAR-2 inhibitor (Figure 1b and Figure 4b), was studied by Cao et al. in 2017. In their randomized, intra-individual parallel-comparative, double-blind, placebo-controlled trial, reduced itch intensity in cowhage-spicules-induced itch was observed in 16 healthy controls after the application of active cream containing MMP versus placebo vehicle cream 30 min after application, demonstrating its promise as an adjunctive agent in chronic pruritic dermatosis [179].

### 2.5. Suppression of the Janus Kinase (JAK)-Signal Transducer and Activator of Transcription (STAT) Pathway, along with the Activation of Suppressor of Cytokine Signaling (SOCS), Is Being Explored

Given their role in the pathogenesis of AD via the Th2 axis immune response, Th17/22 axis immune response, Th1Axis immune response, skin barrier dysfunction, and pruritus in pain, the JAK/STAT pathway has been the target of numerous therapeutic agents for AD [180]. There is a growing preference for selective JAK/STAT inhibitors in current research, with a focus on second-generation inhibitors like upadacitinib and abrocitinib. The first-generation JAK inhibitors, such as ruxolitinib (JAK1/2 inhibitor), baricitinib (JAK1/2 inhibitor), delgocitinib, and tofacitinib (JAK1/2/3 inhibitor), are generally less selective due to structural similarities in the ATP binding site shared among various JAKs [181]. The action of JAK inhibitors is summarized in Figure 3b.

#### 2.5.1. JAK Inhibitors

##### Topical JAK Inhibitors

After demonstrating both safety and efficacy in multiple clinical trials, ruxolitinib 1.5% cream, an inhibitor of JAK1 and JAK2 (Figure 3b), received approval from the US-FDA in September 2021 for the treatment of mild-to-moderate AD in adolescent patients (≥12 years) and adults [182,183].

In a phase IIa study conducted by Bisonnette et al., the topical (ointment) form of tofacitinib, an inhibitor of JAK1, JAK2, and JAK3 (Figure 3b), demonstrated both safety and efficacy in treating mild-to-moderate AD. The study involved 69 adult patients who were randomly assigned to receive either tofacitinib 2% ointment or a placebo. The treatment was applied twice daily for a duration of 4 weeks. The results showed that tofacitinib exhibited a greater mean percentage change in EASI from baseline compared to the placebo group across all endpoints. Based on these findings, tofacitinib shows potential as a therapeutic target for AD [184].

A pan-JAK inhibitor, topical delgocitinib (JTE-052) (Figure 3b), improved the modified EASI (MEASI) scores compared with the vehicle group. This newer topical agent for AD in the formulations of 0.25% ointment for patients aged 2–15 and 0.5% ointment formulations for >16 years has been approved by the Japan FDA [185]. In a meta-analysis by Tsai et al., while topical delgocitinib had a higher rate of achieving an EASI-75 response, other topical agents like ruxolitinib or tofacitinib had demonstrated a better response with regard to IGA and the pruritus-NRS response [186]. A phase III, open-label, and long-term study demonstrated that delgocitinib in both formulations (0.025% and 0.5%) applied twice a day on skin lesions are well tolerated and effective for up to 52 weeks when applied to Japanese infants with AD [187].

Brepocitinib, a selective JAK-1 and TYK-2 inhibitor (Figure 3b), was able to achieve statistically significant reductions from baseline in EASI total score at week 6 compared with their respective vehicles in 292 adolescent (>12 years old) and adult participants with mild to moderate AD at week 6 versus the vehicle treatment in a phase 2b double-blind, dose-ranging study by Landis et al. [188].

Ifidacitinib (ATI-502), a topical JAK inhibitor that selectively targets JAK1 and JAK3 (Figure 3b), has emerged as a promising treatment option for AD. In a phase 2a open-label study conducted by Stacy et al., ifidacitinib applied twice a day for a duration of 4 weeks exhibited favorable results in terms of tolerability, efficacy, and safety. The study involved 17 adult patients with moderate to severe AD. Notably, significant improvements were observed in various assessments, including EASI, pruritus assessment by the subjects, and physician’s global assessment score. The positive effects of the twice-daily application of ATI-502 were evident as early as week 1 and were maintained throughout the 4-week treatment period [189].

##### Oral JAK Inhibitors

Oral tofacitinib (Figure 3b) was used by Levy et al. in an open-label study on patients with moderate to severe AD. Improvements in pruritus score, sleep loss score, and SCORAD index were noted with tofactinib 5 mg once or twice daily in addition to topical treatment for 29 weeks. In their study, the drug was tolerated, with no adverse effects such as infections, cytopenias, transaminitis, decreased renal function, or elevated lipid levels [190]. Oral tofacitinib is a promising drug; however, recently, the FDA issued a warning black box due to the increased risk of serious cardiovascular problems [191].

The European Medicines Agency (EMA) has granted approval for the use of baricitinib, an oral selective Janus kinase 1 (JAK1) and Janus kinase 2 (JAK 2) inhibitor (Figure 3b), in adult patients with AD who are suitable candidates for biologics. The effectiveness and safety of baricitinib in combination with topical corticosteroids were evaluated in the BREEZE-AD4 study, which was a multicenter, double-blind, randomized, placebo-controlled clinical trial. This trial specifically focused on patients with moderate-to-severe AD who had an inadequate response, intolerance, or contraindication to cyclosporine. The study successfully demonstrated the superiority of baricitinib at a dosage of 4 mg in conjunction with topical corticosteroids compared to placebo plus topical corticosteroids. The primary endpoint of achieving at least a 75% improvement in EASI score at week 16 was reached. Additionally, the baricitinib 4 mg group outperformed the placebo, as well as the baricitinib 1 mg and 2 mg groups, in key secondary endpoints such as a ≥4-point improvement in the itch numeric rating scale and the mean change in night-time awakenings. Furthermore, all the groups receiving baricitinib (at doses of 1 mg, 2 mg, and 4 mg) maintained statistically significant improvements compared to the placebo in terms of the percentage change from baseline in EASI score until week 52. While not reaching statistical significance, an important finding in the study was that the number of days without the use of topical corticosteroids was greater in the baricitinib group compared to the placebo group up until week 52 [192]. Recently, baricitinib 4 mg once a day demonstrated its safety and efficacy in patients aged 2 to 18 years of age with moderate-to-severe AD who are candidates for systemic therapy in the BREEZE-ADPEDS study by Torrelo et al. While baricitinib is a promising drug for pediatric patients, this drug might cause effects on bone growth. The BREEZE-AD PEDS study showed that growth parameters continued to increase up to 16 weeks, but monitoring will continue for up to 4 additional years for further classification of the benefit–risk profile [193]. Some of the adverse effects were limited to headache, nasopharyngitis, herpes simplex, and increased creatine phosphokinase.

The US FDA has granted approval for upadacitinib, an oral reversible JAK inhibitor with higher potency for JAK1 (Figure 3b), available in 15 mg and 30 mg doses, as well as Abrocitinib, a selective JAK1 inhibitor (Figure 3b) available in 100 mg and 200 mg doses, for the treatment of AD. The goal for these drugs is to limit the blocking of cytokine axes involved for a much safer profile. A 24-week, head-to-head, phase 3b, multicenter, randomized, double-blinded, double-dummy, active-controlled clinical trial comparing the safety and efficacy of upadacitinib 30 mg daily with subcutaneous dupilumab 300 mg every other week among 692 adults with moderate-to-severe AD who were candidates for systemic therapy demonstrated the superiority of upadacitinib over dupilumab. A total of 71% of patients receiving upadacitinib compared with 61.1% of patients receiving dupilumab achieved EASI-75 at week 16. EASI75 was achieved as early as week 2 (43.7% in upadacitinib versus 17.4% in dupilumab). In addition to their study, patients receiving upadacitinib experienced a decrease in the mean worst pruritus numeral rating scale as early as week 1 [181]. The most common adverse events are upper respiratory tract infection, AD worsening, and acne, but these were not dosage-related [194].

In the JADE COMPARE study—a randomized, double-blind, double-dummy, placebo-controlled trial—a total of 837 adult patients (aged ≥18 years) with moderate-to-severe AD were enrolled. The patients were randomly assigned to receive different treatment regimens for a duration of 16 weeks. The treatment options included oral abrocitinib at doses of 200 mg or 100 mg once daily, subcutaneous dupilumab at a dose of 300 mg every other week (after an initial loading dose of 600 mg), or placebo, in addition to medicated topical therapy. The findings of the study revealed that oral abrocitinib at a dosage of 200 mg demonstrated superiority in treating pruritus compared to subcutaneous dupilumab at a dosage of 300 mg by the end of week 16. Moreover, a greater proportion of patients achieved a ≥4-point improvement from baseline in the Peak Pruritus Numeric Rating Scale (PP-NRS4) with abrocitinib 200 mg, starting as early as day 4 of treatment. Abrocitinib 100 mg exhibited a similar trend, with an improvement in PP-NRS4 observed by day 9. Additionally, significant skin clearance was observed in patients treated with abrocitinib at week 12 of the study [195]. Rapid and effective treatment of AD remains an unmet need, particularly in the difficult-to-treat areas such as the head and neck. A post hoc analysis of JADE COMPARE trial revealed a rapid and persistent improvement in AD with abrocitinib 200 mg reaching EASI-75 at a median of 29 days across body regions, including the head and neck region [196]. In a study by Pavel et al., a significant cellular and molecular suppression of a key AD inflammatory pathway was achieved.

Gusacitinib (ASN002), a dual JAK-SYK inhibitor (Figure 3b), exhibited a broadened spectrum of targeted cytokines (Th2, Th17/Th22, and Th1 pathways) in AD [197]. This broader type of selective inhibition makes it suitable across most of the subtypes of AD as both pediatric [198] and Asian adults [199] have strong Th2 and Th17 polarization, while AD in European American adults has mainly Th2/Th22 polarization [200]. In a recent randomized, double-blind, placebo-controlled trial conducted by Bissonnette et al., the administration of both 40 mg and 80 mg doses of ASN002 once daily for a duration of 28 days demonstrated favorable efficacy and was well tolerated in the treatment of 36 adult patients with moderate-to-severe AD. ASN002 exhibited superiority over the placebo in terms of the proportion of patients achieving EASI-50, EASI-75, and the change from baseline in pruritus (itching) [201].

Zhao et al. conducted a phase II randomized controlled trial in China to evaluate the efficacy and safety of oral ivarmacitinib (SHR0302), a highly selective JAK-1 inhibitor with a mild JAK2-inhibiting effect (Figure 3b) in patients with moderate to severe AD. The trial included 105 adult participants, and after 12 weeks of oral administration, significant improvements in clinical symptoms and skin lesions were observed. The study compared the effects of SHR0302 at doses of 4 mg and 8 mg, as well as a placebo administered once daily. Results showed that both doses of SHR0302 led to notable improvements compared to the placebo. Notably, the SHR0302 8 mg group demonstrated greater efficacy than the SHR0302 4 mg group. At week 12, an IGA response was achieved in 25.7% of the SHR0302 4 mg group, 54.3% of the SHR0302 8 mg group, and 5.7% of the placebo group. Additionally, EASI-75 was achieved in 51.4%, 74.3%, and 22.9% of participants in the SHR0302 4 mg, SHR0302 8 mg, and placebo groups, respectively. These findings demonstrate the positive impact of oral administration of SHR0302 in treating moderate-to-severe AD [202].

Jaktinib hydrochloride is a pan-JAK inhibitor (Figure 3b) that can suppress JAKl, JAK2, JAK3, and TYK2 at the cellular level, further blocking the JAK-STAT signaling pathways and significantly relieving the inflammation due to immune reactions initiated by cytokines including IL-2, IL-4, IL-6, IL-7, and IL-10. There is potential for jaktinib hydrochloride to treat cytokine storms in patients with COVID-19. Jaktinib in oral form was well tolerated in a single dose ranging from 25 to 400 mg and multiple doses up to 200 mg every 24 h, in a phase I, randomized, double-blind, placebo-controlled, single-ascending-dose, multiple-ascending-dose, and food effect study to evaluate its tolerance in healthy Chinese volunteers [203]. A multi-center, randomized, double-blind, placebo, parallel-controlled phase II clinical study of jaktinib hydrochloride tablets in the treatment of moderate and severe AD patients completed their recruitment as of March 2023. Their study was divided into two stages: the first stage test (1–12 weeks): the main test; and the second stage test (13–24 weeks): the extended test. The trial set up for the main test was composed of 4 treatment groups, including 3 dose exploration groups: Jaktinib 50 mg twice a day, 75 mg twice a day, and 100 mg twice a day groups, and 1 placebo control group. The extended test was composed of randomly assigned subjects in the placebo group to receive Jaktinib 50 mg twice a day, 75 mg twice a day, and 100 mg twice a day groups at a ratio of 1:1:1 (randomized in a blinded state and completed by IWRS in the background), respectively, while the main test group (receive Jaktinib 50 mg twice a day, 75 mg twice a day, 100 mg twice a day group) subjects still maintained the original dose after 12 weeks [204]. However, as of this writing, no publications are yet available. A multicenter, randomized, double-blind, placebo-controlled phase III clinical study of jaktinib hydrochloride tablets in the treatment of adult patients with moderate and severe AD is currently underway and is expected to be completed in September 2023 [205]

The European Medicines Agency recommended the use of JAK inhibitors such as abrocitinib, filgotinib, baricitinib, and tofacitinib in caution in patients with risk of cardiovascular problems, malignancy, venous thromboembolism, and serious infections. In addition, the committee also advises the use only JAK inhibitors if no other suitable treatment is available.

Oral JAK inhibitors are a promising group of drugs that may provide a solution to the unmet need for patients who are intolerant or unresponsive to systemic agents or for patients with needle phobia.

#### 2.5.2. Suppressor of Cytokine Signaling (SOCS)

The family of suppressors of cytokine signaling (SOCS) comprises several members, including SOCS-1, SOCS-2, SOCS-3, SOCS-4, SOCS-5, SOCS-6, SOCS-7, and cytokine-inducible SH2 protein (CIS). As their name suggests, these molecules act as negative feedback stimulators, targeting the downstream components of the JAK-STAT pathway. In the context of AD pathophysiology, the role of SOCS-1, SOCS-3, and SOCS-5 as negative modulators is particularly significant. The modulation of the JAK/STAT/SOCS pathway using natural biomolecules may hold potential as a therapeutic target for future drug development.

##### Suppressor of Cytokine Signaling-1 (SOCS-1)

In vitro, in vivo, and human studies have shown that SOCS-1 inhibits IFN-γ signaling by binding to JAK1 and JAK2. This results in the inhibition of STAT1 phosphorylation of STAT1. Disabled Jak1 and Jak2 cannot mediate STAT1 phosphorylation, which is necessary for the activation of γ-activated sequence (GAS) inflammatory genes [206]. A recent study by Coelho et al. reported that SOCS-1 controls inflammation by specifically targeting the p65 nuclear factor-κb (NF-κB) in inactivated macrophages, thereby suppressing the pro-inflammatory transcription [207]. A short motif called the Kinase Inhibitory Region (KIR) was found in SOCS-1 and, through its direct binding with JAK2, this interaction was able to inhibit the tyrosine kinase activity of JAK2 [208].

##### Suppressor of Cytokine Signaling-3 (SOCS-3)

Increased expression of SOCS-3 was observed in the skin of AD patients than in healthy individuals. In a mouse model, the overexpression of the SOCS-3 protein impaired Th1 differentiation and elevated levels of Th2 responses [209,210]. SOCS-3 inhibited the IL-6 pathway through STAT3 activation [211] and inhibited IL-12 through STAT4 activation. The results of the molecular studies performed by Babon et al. showed that SOCS-3 directly inhibited JAK1, JAK2, and TYK2 but did not inhibit JAK3 [212]. SOCS-3 also contains a Kinase Inhibitory Region (KIR), which competitively inhibits the activity JAK kinase. The role of SOCS-3 in the Th2-mediated allergic response might be a potential target in the treatment of AD.

##### Suppressor of Cytokine Signaling-5 (SOCS-5)

While little is known about the function of SOCS-5, it has been shown to be involved in a variety of allergic disease states such as AD and asthma. Seki et al. concluded that SOCS-5 inhibits IL-4-dependent signaling toward the control of Th2 differentiation. In the mouse study of Sharma et al., the group was able to demonstrate the role of SOCS-5 in regulating the activation of STAT proteins and the phosphorylation of JAK1/2/3 but not TYK2 [213]. SOCS protein mimetics found in various natural biomolecules are currently being developed in vitro and may provide novel insights in the regulation of Th2 and Th1 immune responses.

Curcumin from Curcuma longa in various experimental studies demonstrated its ability to upregulate SOCS1 and inhibit phosphorylation of JAK2, STAT3, and STAT6 [214,215,216]. In an ovalbumin-induced AD, Sharma et al. was able to show the suppression of Th2-promoting cytokines (TSLP/IL-33) and Th2 cytokines (IL-4/IL-5/IL-13/IL-31), along with decreased STAT6 phosphorylation [217]. In an in vitro study conducted by Zhang et al., resveratrol, a polyphenol and phytoalexin antioxidant present in various plants, was found to enhance the expression of SOCS-1 and inhibit JAK-STAT signaling [218]. Ma et al. in their in vivo study showed that resveratrol suppressed the production of cytokines in LPS-stimulated RAW264.7 cells via the upregulation of SOCS-1 expression and inhibition of STAT1/STAT3 [219]. Luteolin is a flavone found in extracts of *Reseda luteola* or can be taken as a dietary source in various fruits such as carrots, apple skins, peppers, rosemary, and thyme. This flavone was able to demonstrate an increased expression of SOCS-3 and suppressed STAT1/STAT3 phosphorylation in in vitro studies [220]. Furthermore, in a mouse study by Baolin et al., topical application of luteolin inhibited the scratching behavior associated with allergic cutaneous reactions [221]. Leunorine, an alkaloid extracted from *Herba leunori*, increased SOCS-5 expression and decreased the expression of JAK2/STAT3 in the in vitro studies of Liu et al. [222]. Astragalin, a flavonoid found in extracts of persimmon and *Rosa agrestis* leaves, showed an elevated SOCS-5 expression, decreased Th2 differentiation, and downregulation of IL-4, IL-5, IL-15, IL-31, and IgE in mouse models [223,224].

### 2.6. The Hippo-YAP Pathway

It is established that the Th1/Th2 imbalance is involved in the pathophysiology of AD. However, recent studies have shown that the Th17 and Treg cells play important roles in AD [225]. Yes-associated protein (YAP) is a key downstream member of Hippo signaling and may be an essential factor for the maintenance of Th17/Treg cell equilibrium. The importance of the Hippo-YAP pathway and its role in Th17 and Treg cell differentiation has been cited in numerous studies. In the study of Xia et al., 35 patients with AD and 24 healthy controls were enrolled. Peripheral venous blood was collected from patients and healthy controls to isolate serum or separate the peripheral blood mononuclear cells. Also in their study, an AD mouse model was constructed using 2,4-dinitrofluorobenzene, and an AD-like inflammatory cell model was constructed using TNF-α/IFN-**γ**-activated HaCaT cells to detect Th1/Th2/Th17/Treg cell imbalances using flow cytometry. The group found out that a high expression of YAP was found in healthy individuals and mice, suggesting its role in the function and maintenance of Tregs [226]. Their study was similar to that of Ni et al., where the isolated Treg cells from the peripheral blood of healthy humans or mice spleen expressed a high expression of YAP compared with Th1, Th2, and Th17 cells [227]. AD being a state of constant inflammation may be a result of the downregulation of YAP expression in Tregs. In addition to their findings, an increased expression of YAP was seen in Th17 cells and was more apparent in acute AD. In the presence of IL-2 and GF-b, CD4+ CD25+ Tregs can induce the expression of FOXP4 and maintain the number of peripheral CD4+ CD25+ Tregs. While in the presence of IL-6, TGF-B promotes the naïve T cells to differentiate to Th17 cells [226]. In the study of Schlegelmilch et al., the group was able to conclude that YAP acts as a critical modulator of epidermal stem cell proliferation and tissue expansion through interaction with transcriptional enhanced associate domain (TEAD) transcription factors [228] and it plays a crucial role in the development of a thick three-dimensional structure of the epidermis [229]. The downregulation of YAP in the epidermis may contribute to the development of skin erosion due to the delayed cell proliferation and migration and increased apoptosis of keratinocytes, thus influencing the impairment of skin barrier function of AD [226]. Further studies are needed to determine the exact role of YAP in skin development and maintenance. A pharmaceutical intervention targeting the expression of YAP in the different cells may provide a newer treatment option in targeting the pathophysiology of AD.

## 3. Conclusions

In our review article, we focused on the challenges and the recently developed pharmacological agents in AD based on the specific pathophysiology through modulation of the skin microbiome and improvement of the skin barrier in combination with the selection of appropriate agents that target the innate and adaptive immune system that would significantly make an impact in improving the itch–scratch cycle.

A new era in the treatment of AD treatment is approaching, and much more effective therapeutic targets based on molecular and therapeutic phenotype selection will be more readily available and, as a result, every individual patient will have tailored and more personalized treatment options. Recent findings of various clinical trials also showed that we might have opportunities to use a blend or combination of different drugs if there is an insufficient response.

Biologic agents will continue to dominate the AD treatment market for years to come. JAK inhibitors offer an important oral option to the AD population. However, more long-term studies are needed on the maintenance of therapeutic efficacy and safety of novel drugs targeting newly discovered cytokines and receptors.

More importantly, a multidisciplinary approach is important for achieving a better health-related quality of life in patients with AD. The importance of patient and caregiver education cannot be overly emphasized. A deeper understanding of the complex pathogenesis of AD will lead to the development of newer therapeutic targets and innovative therapeutics.

## Figures and Tables

**Figure 1 ijms-24-11380-f001:**
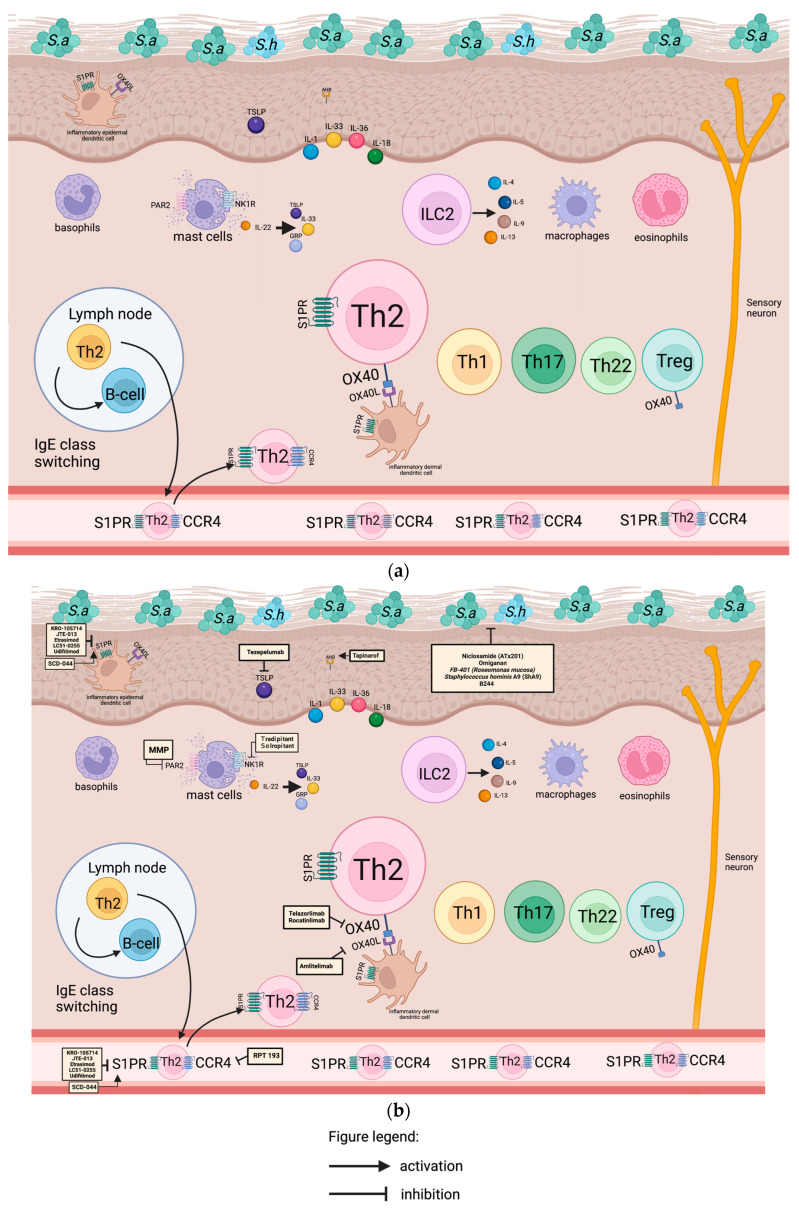
(**a**) Overview of the skin microbiome and innate immune system in atopic dermatitis. Skin-resident dendritic cells are activated by triggers of atopic dermatitis. Once triggered, dendritic cells (DC) migrate to the local lymph nodes to prime the naive T helper cells and shift these cells toward Th2 polarization. IgE class switching to B cells is also induced by Th2 cells. T-helper 2 cells are recruited back to the skin, along with Th1, Th17, and Th22 generating the “immunological march” of atopic dermatitis. Sa, staphylococcus aureus; Sh, staphylococcus hominis; inflammatory epidermal dendritic cell; S1PR, sphingosine 1-phosphate receptor; TSLP, thymic stromal lymphopoietin; AHR, aryl-hydrocarbon receptor; basophils; PAR2, protease-activated receptor 2; NK1R, neurokinin 1 receptor; GRP, gastrin-releasing peptide; ILC2, innate lymphoid cell 2; macrophages; eosinophils; lymph node, inflammatory dermal dendritic cell; Th2: T-helper 2 cell; OX40; OX40L: OX40 ligand; B-cell; Th1, T-helper 1 cell; Th17, T-helper 17 cell; Treg, regulatory T cell; mast cells; sensory neuron; CCR4: C-C motif chemokine receptor 4. This figure was created with Biorender at www.biorender.com. (**b**) Mechanism of action of different drugs acting on the skin microbiome and innate immune system in atopic dermatitis. KRO-105714 is a dual antagonist of sphingosylphosphorylcholine and sphingosine-1-phosphate receptor 1 in topical form. JTE-013 is an S1PR2 antagonist in topical form. Etrasimod, LC51-0255, and udifitimod are oral S1PR modulators. Tezepelumab is an anti-TSLP antibody. Tapinarof is an AHR agonist. Niclosamide, omiganan, FB 401 (*Roseomonas mucosa*), *Staphylococcus hominis* A9, and B244 are modulators of skin microbiome. VLY-686 and selropitant are NK1R antagonists. Telazorlimab (GBR830) and rocatinlimab are humanized monoclonal anti-OX40 antibodies. Amlitelimab is a monoclonal antibody that targets OX40L. RPT 193 is a CCR4 antagonist. Sa, staphylococcus aureus; Sh, staphylococcus hominis; inflammatory epidermal dendritic cell; S1PR, sphingosine 1-phosphate receptor; TSLP, thymic stromal lymphopoietin; AHR, aryl-hydrocarbon receptor; basophils; PAR2, protease-activated receptor 2; NK1R, neurokinin 1 receptor; GRP, gastrin-releasing peptide; ILC2, innate lymphoid cell 2; macrophages; eosinophils; lymph node, inflammatory dermal dendritic cell; Th2: T-helper 2 cell; OX40; OX40L: OX40 ligand; B-cell; Th1, T-helper 1 cell; Th17, T-helper 17 cell; Treg, regulatory T cell; mast cells; sensory neuron; CCR4: C-C motif chemokine receptor 4. This figure was created with Biorender at www.biorender.com.

**Figure 2 ijms-24-11380-f002:**
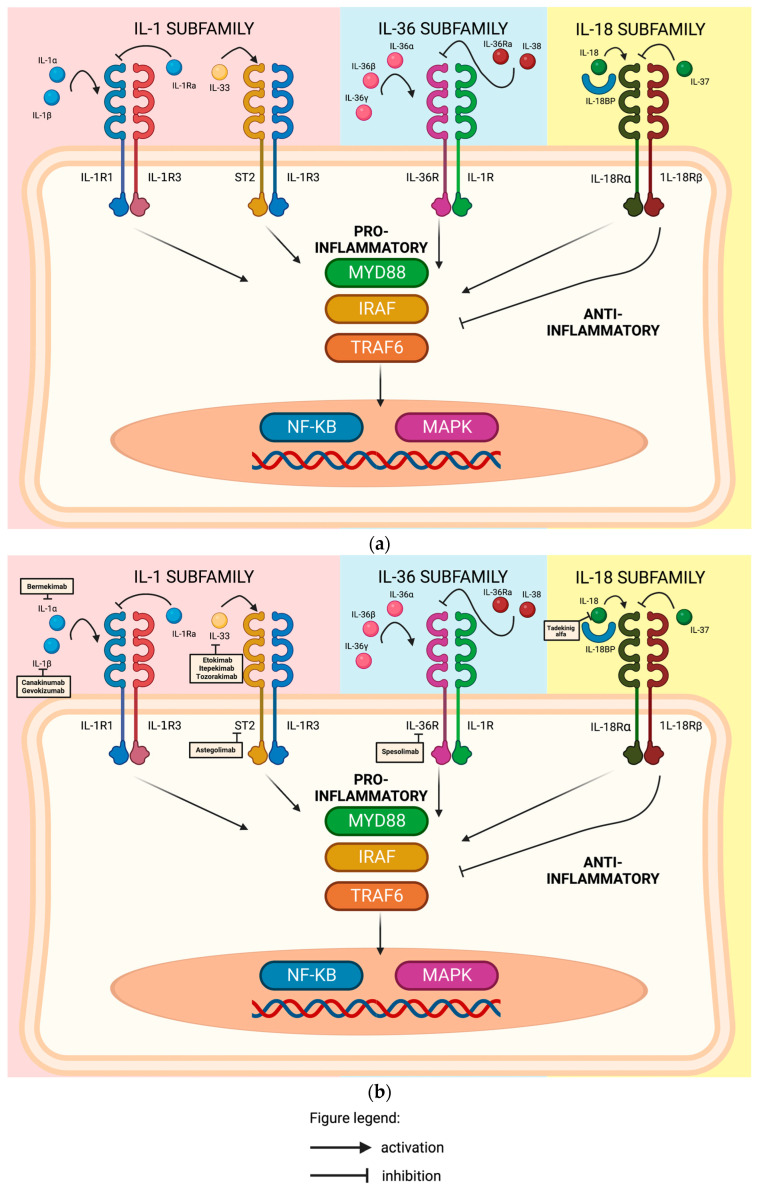
(**a**) Members of the Interleukin-1 (IL-1) family of cytokines and their receptors inside a keratinocyte. To exert its pro-inflammatory effect, Interleukin-1 alpha (IL-1α) and Interleukin-1 beta (IL-1β) bind to Interleukin-1 receptor type 1 (IL-1R1) and recruits Interleukin-1 receptor accessory protein (IL-1R3). Interleukin-1 receptor antagonist (Il-1Ra) is a natural anti-inflammatory protein that suppresses the pro-inflammatory effect of Interleukin-1. Interleukin-33 (IL-33) binds to receptor ST2 and further recruits Interleukin-1 receptor accessory protein (IL-1R3) to exert its pro-inflammatory effect. Interleukin-36 alpha (IL-36α), Interleukin-36 beta (IL-36β), and Interleukin-36 gamma (IL-36γ) bind to Interleukin-36 receptor (IL-36R) and recruit Interleukin-1 receptor accessory protein (IL-1R3) to induce inflammation, while Interleukin-36 receptor antagonist (IL-36Ra) and Interleukin-18 (IL-18) suppress this binding. IL-18 cytokine binds to interleukin-18 receptor alpha (1L18Rα) and recruits interleukine-18 receptor beta (1L18Rβ) to stimulate the production of IFN-γ, inducing a Th2 and Th1 response. Interleukin-18 binding protein (IL18-BP) regulates its pro-inflammatory activity by creating a negative feedback mechanism by sequestering the cytokine Interleukin 37 (IL-37) to bind to interleukin-18 receptor alpha (1L18Rα) and restrict the interleukin-18 receptor (1L-18R)-dependent inflammation and inhibiting its pro-inflammatory cytokine production. The binding of these cytokines to these specific ligands results in activation of NFKB or MAPK signaling pro-inflammatory gene expression through MyD88, IRAF, and/or TRAF6 signaling mechanisms. MyD88, Myeloid differentiation primary response 88; IRAF, interferon regulatory factor; TRAF6, TNF-receptor-associated factor 6; NFKB, Nuclear factor kappa-light-chain-enhancer of activated B cells; MAPK, mitogen-activated protein kinase 11. This figure was created with Biorender at www.biorender.com. (**b**) Mechanism of action of different drugs acting on the members of the Interleukin-1 (IL-1) family of cytokines and their receptors inside a keratinocyte. Bermekimab, a fully human monoclonal antibody that targets IL-1α. Canakinumab, a fully human monoclonal antibody against IL-1β. Gevokizumab, a neutralizing humanized monoclonal antibody specific to IL-1β. Etokimab is a humanized anti-IL-33 monoclonal antibody. Itepekimab (REGN3500) is an anti-IL33 antibody. Tozorakimab (MEDI3506) inhibits IL33 signaling via IL33R/ST2 pathway. Astegolimab, a fully human IgG2 monoclonal antibody that binds to ST2. Spesolimab, an antibody against IL36R. Tadekinig alfa is human recombinant IL-18-binding protein that neutralizes IL-18. MyD88, Myeloid differentiation primary response 88; IRAF, interferon regulatory factor; TRAF6, TNF-receptor-associated factor 6; NFKB, Nuclear factor kappa-light-chain-enhancer of activated B cells; MAPK, mitogen-activated protein kinase 11. This figure was created with Biorender at www.biorender.com.

**Figure 3 ijms-24-11380-f003:**
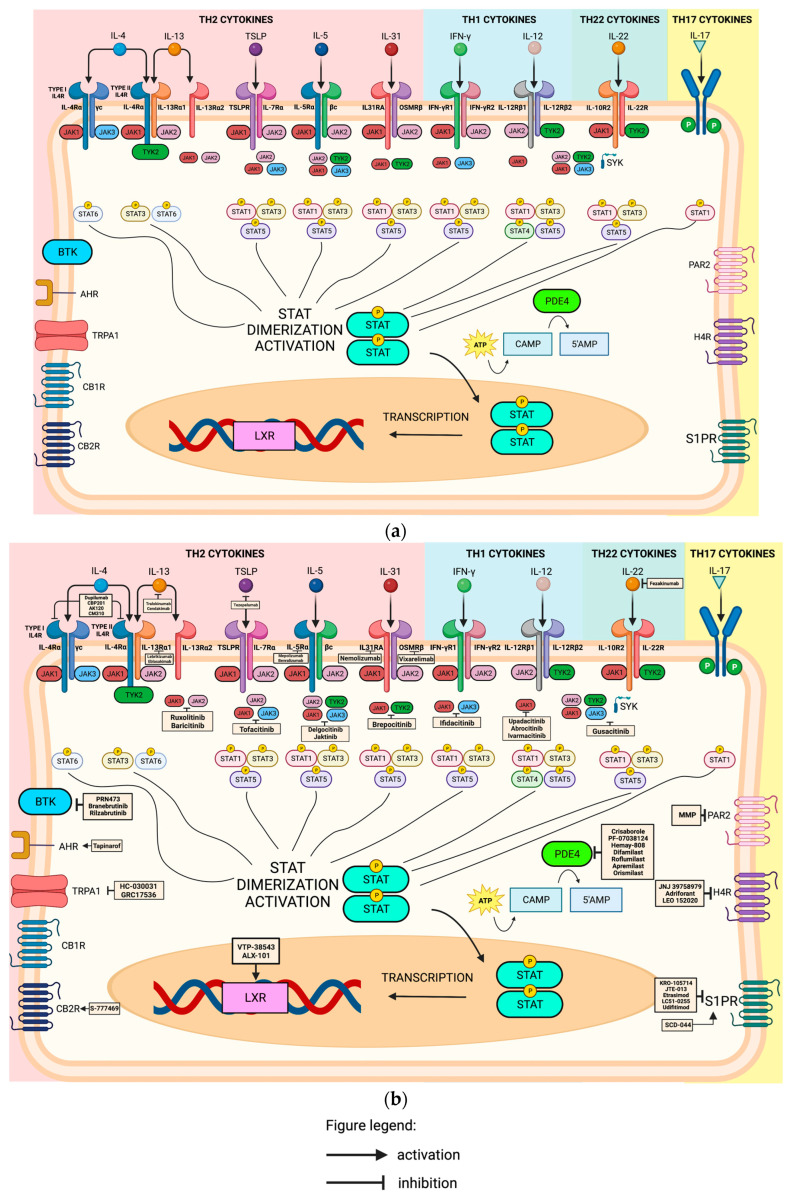
(**a**) Activation of the adaptive immune system via various receptors, different interleukins and its receptors, and the Janus kinases (JAK) STAT pathway. JAK1, Janus kinase 1; JAK2, Janus kinase 2; JAK3, Janus kinase 3; TYK2, Tyrosine kinase 2; STAT1, Signal transducer and activator of transcription 1; STAT3, Signal transducer and activator of transcription 3; STAT4, Signal transducer and activator of transcription 4; STAT5, Signal transducer and activator of transcription 5; STAT6, Signal transducer and activator of transcription 6; SYK, Spleen tyrosine kinase; BTK, Bruton’s tyrosine kinase; AHR, Aryl-hydrocarbon receptor; TRPA1, Transient receptor cation ankyrin 1; CB1R, Cannabinoid type 1 receptor; CB2R, Cannabinoid type 2 receptor; LXR, Liver X receptor; STAT, Signal transducer and activator of transcription; PDE4, Phosphodiesterase-4; PAR2, Protease-activated receptor 2; H4R, Histamine 4 receptor; S1PR, Sphingosine-1-phosphate receptor. This figure was created with Biorender at www.biorender.com. (**b**) Mechanism of action of different drugs acting on the adaptive immune system via various receptors, different interleukins and its receptors, and the Janus kinases (JAK) STAT pathway. Dupilumab is a fully humanized monoclonal antibody targeting IL4Rα receptor. CBP201 is a human IgG4 monoclonal antibody that binds to IL-4Rα receptor. AK120 is an antibody against IL4Rα receptor. Tralokinumab and cendakimab, a fully humanized IgG4λ, bind to IL-13 at an epitope that overlaps with the binding site of receptors IL13Rαl subunit and IL13Rα2 subunit. Tezepelumab is an anti-TSLP antibody. Lebrikizumab, a humanized monoclonal IgG4 antibody that selectively blocks IL13 and prevents heterodimerization of type II IL4R receptor composed of IL4Rα and IL13Rαl subunit. Eblasakimab is a fully humanized monoclonal antibody that targets the IL-13Rα1 subunit, therefore inhibiting the signals by IL4 and IL-13. Mepolizumab, a humanized immunoglobulin G monoclonal antibody, binds to interleukin-5, inhibiting its activity. Benralizumab directly binds to the IL-5Rα chain to inhibit the signaling pathway. Nemolizumab, a humanized monoclonal antibody that antagonizes IL-31RA receptor. Vixarelimab, a first-in-class fully human monoclonal antibody that targets oncostatin M receptor. Fezakinumab is an anti-IL-22 monoclonal antibody. Ruxolitinib, baricitinib, and ivarmacitinib inhibit JAK1 and JAK2. Tofacitinib is an inhibitor of JAK1, JAK2, and JAK3. Delgocitinib and jaktinib are pan-jak inhibitors inhibiting JAK1, JAK2, JAK3, and TYK2. Brepocitinib is a selective JAK-1 and TYK-2 inhibitor. Ifidacitinib selectively targets JAK1 and JAK3. Upadacitinib, abrocotinib, and ivarmacitinib are JAK inhibitors with a higher potency for JAK-1. Gusacitinib is a dual JAK-SYK inhibitor. PRN473 is a covalent BTK inhibitor. Branebrutinib is a highly selective BTK inhibitor that covalently binds to the cysteine residue of BTK. Tapinarof is an AHR agonist. HC-030031 and GRC17536 are both TRPA1 antagonists. S-777469 is a CB2R agonist. VTP-38543 and ALX-101 are LXR-β agonists. Crisaborole, PF-07038124, Hemay-808, difamilast, and roflumilast are PDE4 inhibitors in topical forms, while apremilast and orismilast are PDE4 inhibitors in oral form. Methylbenzyl methylbenzimidazole piperidinyl methanone (MMP) is a selective PAR-2 inhibitor in topical form. JNJ 39758979, adriforant, and LEO 152020 are H4R antagonists. KRO-105714 is a dual antagonist of sphingosylphosphorylcholine and sphingosine-1-phosphate receptor 1 in topical form. JTE-013 is an S1PR2 antagonist in topical form. Etrasimod, LC51-0255, and udifitimod are oral S1PR modulators. SCD-044 is an S1PR1 agonist. JAK1, Janus kinase 1; JAK2, Janus kinase 2; JAK3, Janus kinase 3; TYK2, Tyrosine kinase 2; STAT1, Signal transducer and activator of transcription 1; STAT3, Signal transducer and activator of transcription 3; STAT4, Signal transducer and activator of transcription 4; STAT5, Signal transducer and activator of transcription 5; STAT6, Signal transducer and activator of transcription 6; SYK, Spleen tyrosine kinase; BTK, Bruton’s tyrosine kinase; AHR, Aryl-hydrocarbon receptor; TRPA1, Transient receptor cation ankyrin 1; CB1R, Cannabinoid type 1 receptor; CB2R, Cannabinoid type 2 receptor; LXR, Liver X receptor; STAT, Signal transducer and activator of transcription; PDE4, Phosphodiesterase-4; PAR2, Protease-activated receptor 2; H4R, Histamine 4 receptor; S1PR, Sphingosine-1-phosphate receptor. This figure was created with Biorender at www.biorender.com.

**Figure 4 ijms-24-11380-f004:**
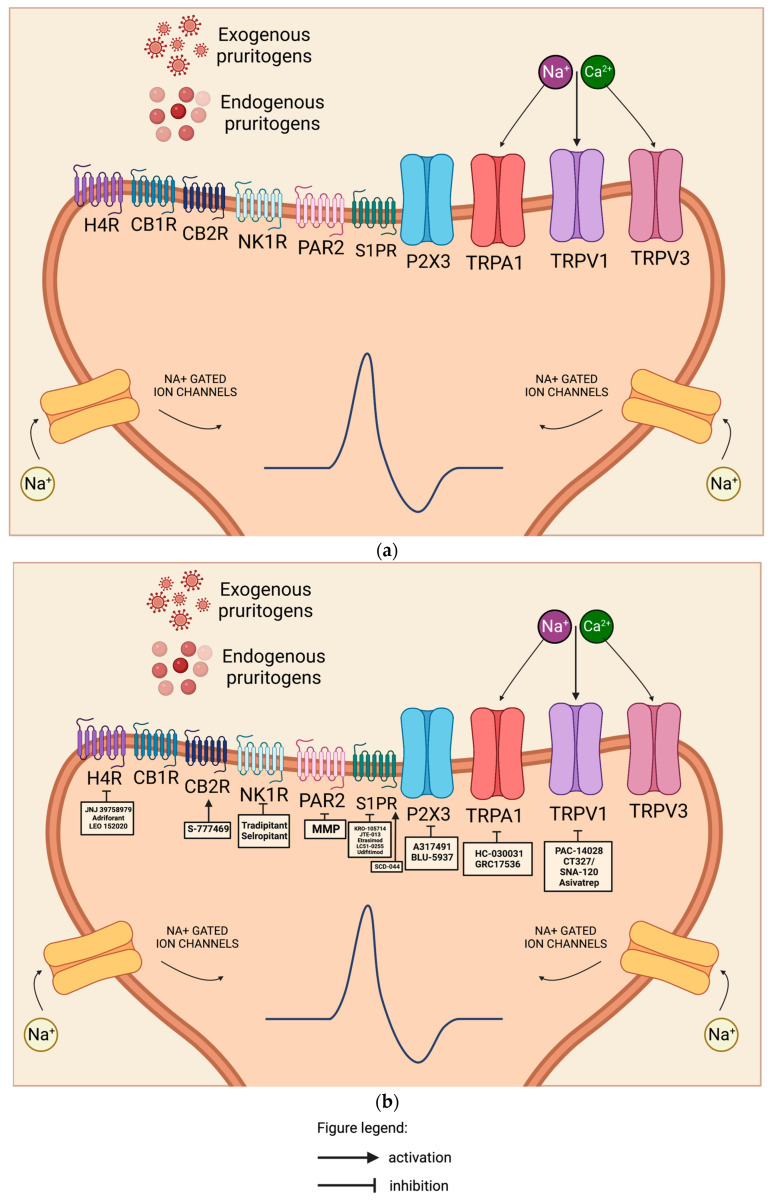
(**a**) Pyschosomatic aspect of the itch–scratch cycle in atopic dermatitis. This cycle disrupts the epidermal barrier, causes damage to keratinocytes, activates local dendritic cells, and triggers the adaptive response. Exogenous pruritogens like allergens, pathogens, toxins, and irritants, as well as endogenous pruritogens like cytokines, neuropeptides, signaling lipids, and proteases released from keratinocytes and immune cells, bind to receptors on cutaneous itch sensory neurons. These sensory neurons, primarily composed of unmyelinated C-fibers, transmit the itch sensation as action potentials to specific dorsal root ganglia and the central nervous system. The binding of mediators to their respective receptors initiates a signaling cascade involving secondary messengers, which activate TRP family cation channels (particularly TRPA and TRPV), ultimately leading to the opening of voltage-gated sodium channels and neuronal depolarization. Na^+^, sodium ion; Ca^+^, calcium ion; H4R, Histamine 4 receptor; CB1R, Cannabinoid type 1 receptor; CB2R, Cannabinoid type 2 receptor; NK1R, neurokinin 1 receptor; PAR2, Protease-activated receptor 2; S1PR, Sphingosine-1-phosphate receptor; P2X, purinoreceptors 3 (P2XR3); TRPA1, Transient receptor cation ankyrin 1; TRPV1, Transient receptor potential channel vanilloid 1; TRPV3, Transient receptor potential channel vanilloid 3. This figure was created with Biorender at www.biorender.com. (**b**) Mechanism of action of different drugs acting on various receptors expressed in the sensory neuron responsible for the pruritus in atopic dermatitis. JNJ 39758979, adriforant, and LEO 152020 are H4R antagonists. S-777469 is a CB2R agonist. Tradipitant and selropitant are NK1R antagonists. Methylbenzyl methylbenzimidazole piperidinyl methanone (MMP) is a selective PAR-2 inhibitor. KRO-105714 is a dual antagonist of sphingosylphosphorylcholine and sphingosine-1-phosphate receptor 1 in topical form. JTE-013 is an S1PR2 antagonist in topical form. Etrasimod is an oral S1PR modulator. SCD-044 is an S1PR1 agonist. A317491 and BLU-5937 are selective P2XR3 antagonists. HC-030031 is a topical TRPA1 antagonist while GRC17536 is an oral TRPA1 antagonist. PAC-14028, CT327/SNA-120, and asivatrep are all TRPV1 antagonists. Na^+^, sodium ion; Ca^+^, calcium ion; H4R, Histamine 4 receptor; CB1R, Cannabinoid type 1 receptor; CB2R, Cannabinoid type 2 receptor; NK1R, neurokinin 1 receptor; PAR2, Protease-activated receptor 2; S1PR, Sphingosine-1-phosphate receptor; P2X purinoreceptors 3, P2XR3; TRPA1, Transient receptor cation ankyrin 1; TRPV1, Transient receptor potential channel vanilloid 1; TRPV3, Transient receptor potential channel vanilloid 3. This figure was created with Biorender at www.biorender.com.

## Data Availability

Not applicable.

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
