# Peer review of "Challenges and Future Trends in Atopic Dermatitis"

_ijms, 2023, doi:10.3390/ijms241411380_

Round 1

Reviewer 1 Report

Atopic dermatitis is a complex, multidimensional and interactive disease that presents 10 potential fields of preventive and therapeutic management. In addition to the treatment armamentarium available for atopic dermatitis, novel drugs targeting significant molecular pathways in atopic dermatitis biologics and small molecules are also being developed given the condition’s complex pathophysiology. While most of the patients are expecting better efficacy and long-term control, response to these drugs would still be up to numerous factors such as complex genotype, diverse environmental triggers and microbiome derived signal and most importantly dynamic immune response. This review article will highlight the challenges and the recently developed pharmacological agents in atopic dermatitis based on the molecular pathogenesis of this condition creating a specific therapeutic approach towards a more personalized medicine

My opinion

The article is organised and well-written.

It is interesting and gives new information on a a novel trends in atopic dermatitis.

It evaluates the newest literature and gives an up-to-date information on the new approaches in the AD treatment.

Figures give the additional value to the text of the manuscript.

Minor revision of the english language should be done.

Reviewer 2 Report

First of all, congratulations to the authors for the hard work collecting all those information.

-The manuscript should be edited for correct use of English language. There are many mistakes in spelling, use of prepositions, tenses etc. For example, (line 10) what is an “interactive” disease?,  (line 27) based ON… , (line 28) various differenceS (plural), (line 48) SUCH AS biologics and…, (line 67) is incomprehensible, maybe do it “this review article is divided in 5 sub-categories”, (line 312) anti-IL33 antibody (singular), (line 340) are you sure for the word “promiscuous”?,(line 701) pde4 IS expressed, not ARE,(line 799) histamines CAUSE, not causes, (line 827) in patients WITH, (line 849) the inhibiting OF, (line 859) and IS estimated to be completed BY, (line 923) IN august 2021, since the last august is that of 2022 and soon that of 2023, etc. etc.

-Line 52: no need to mention all drugs again, they are all in line 44

-Line 172: you mention 4 antagonists and there are only 3 in the parenthesis.

-Line 193: which studies?

-Line 214: the target is the interleukin, not the antibody.

-Line 638: what is ASLAN? The code name of eblasakimab?

-Line 1003: was it a thesis? It looks like a normal article.

-There are some acronyms in the text that are not explained and some that should be abbreviated, for example in line 894 you don’t need to write “sphingosine-1-phosphate receptor 1” again, it is explained in line 869.

-There are countless “Figure a” or “Figure b” in the text, without the number of the figure.

-Since the article is very long and it concerns future treatments, I suggest making lines 465-504 and 557-589 very shorter (2-3 lines), like you did with ruxolitinib, since dupilumab and tralokinumab are already approved and used.

-To add more info for the readers, I suggest citing:

 10.1016/j.jdermsci.2019.08.006” in the IL-33 general information part (lines 286-302)

“10.3390/jcm11174974”, which has much information about JAKs, at line 1273 (together with reference 188)

Extensive editing of English language required

Author Response

Please see the attachment. Thank you very much for the comments.

Round 2

Reviewer 2 Report

-

-